# G2D2: Gradient-guided Discrete Diffusion for image inverse problem solving

## Abstract

Recent literature has effectively leveraged diffusion models trained on continuous variables as priors for solving inverse problems. Notably, discrete diffusion models with discrete latent codes have shown strong performance, particularly in modalities suited for discrete compressed representations, such as image and motion generation. However, their discrete and non-differentiable nature has limited their application to inverse problems formulated in continuous spaces. This paper presents a novel method for addressing linear inverse problems by leveraging image-generation models based on discrete diffusion as priors. We overcome these limitations by approximating the true posterior distribution with a variational distribution constructed from categorical distributions and continuous relaxation techniques. Furthermore, we employ a star-shaped noise process to mitigate the drawbacks of traditional discrete diffusion models with absorbing states, demonstrating that our method performs comparably to continuous diffusion techniques. To the best of our knowledge, this is the first approach to use discrete diffusion model-based priors for solving image inverse problems.

## 1 Introduction

Diffusion models have gained significant attention as deep generative models, achieving remarkable success in image (Sohl-Dickstein et al., 2015; Ho et al., 2020; Song et al., 2021b; Dhariwal & Nichol, 2021; Esser et al., 2024), audio (Liu et al., 2023; Chen et al., 2024a), and video generation (Ho et al., 2022b;a). These models operate by iteratively corrupting data then learning to reverse this corruption process, ultimately generating high-quality samples from noise. In parallel with continuous diffusion models, discrete diffusion models have emerged as a compelling alternative. These models have gained traction by demonstrating notable results not only in image (Gu et al., 2022), audio (Yang et al., 2023), and text generation (Austin et al., 2021; Lou et al., 2023a) but also in more specialized areas such as motion data (Lou et al., 2023b; Pinyoanuntapong et al., 2024), protein synthesis (Gruver et al., 2024), and graph generation (Vignac et al., 2023).

Building on these advancements, researchers have made significant progress in expanding the application of diffusion models. They have explored using diffusion models, trained either directly on pixel space or on latent representations derived from variational autoencoders (VAEs), to address inverse problems (Kawar et al., 2022; Chung et al., 2023b; Wang et al., 2023) and carry out various conditional-generation tasks (Yu et al., 2023; Bansal et al., 2024; He et al., 2024) without the need for additional training. These efforts aim to use the powerful generative capabilities of diffusion models to tackle intricate problems and generate conditional outputs, all while preserving the models' original trained parameters.

This line of work has been primarily restricted to diffusion models trained in continuous spaces, and methods using trained discrete diffusion models as priors remain limited Gruver et al. (2024); Chen et al. (2024b); Li et al. (2024). The inherent nature of the generation process in discrete diffusion models involves non-differentiable operations, posing a challenge for their application to inverse problems formulated in continuous spaces. Therefore, controlling discrete diffusion models often necessitates an additional trained network (Gruver et al., 2024; Nisonoff et al., 2024; Klarner et al., 2024; Vignac et al., 2023). Training-free methods have been confined to relatively low-dimensional data (Chen et al., 2024b) or to specific tasks such as image inpainting (Gu et al., 2022).

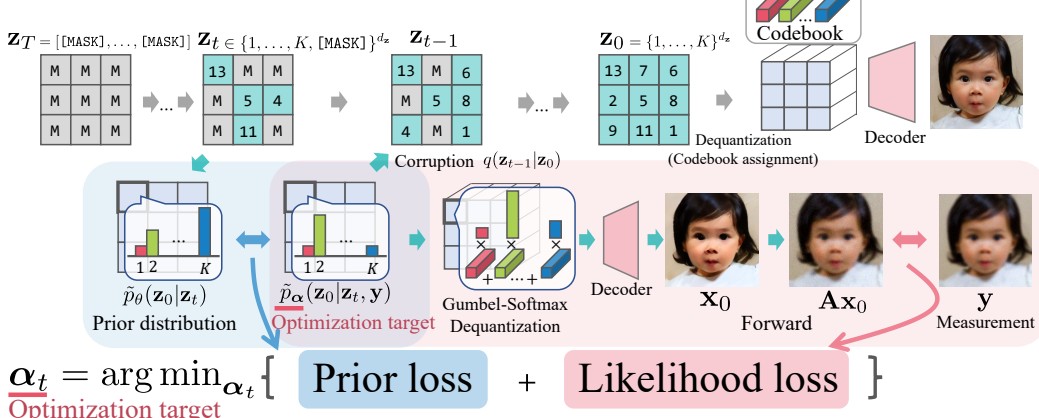

Figure 1: **Illustration of G2D2**. At each time step $t$, variational categorical distribution $\tilde{p}_\alpha$ is optimized with respect to sum of prior loss and likelihood loss, followed by sampling $\mathbf{z}_{t-1}$. Both terms are continuously differentiable, enabling continuous optimization.

This limitation constrains the utilization of the powerful generative capabilities in domains such as motion data, where generative models with discrete latent spaces have demonstrated remarkable success. This constraint motivates the exploration of discrete diffusion models as priors, given their potential advantages in representing complex data distributions and generating high-fidelity samples. To the best of our knowledge, in the context of inverse problems for motion data and image data, there exists no prior work that uses discrete diffusion models as priors.

Controlling the generation of diffusion models without additional training generally involves manipulating the generation trajectory using gradients in continuous space. The primary objective here is to generate samples from the prior model that have high likelihood with respect to the measurement equation of the inverse problem or the guidance target. A specific approach to achieve this involves adjusting samples during the diffusion model's generation process by using gradients of a loss function that computes the likelihood. This methodology has been demonstrated in the works of Chung et al. (2023b) and Yu et al. (2023).

We propose Gradient-guided Discrete Diffusion (**G2D2**), an inverse problem solving method that overcomes the aforementioned limitations while using a discrete diffusion model as a prior. Our focus is on solving image inverse problems using a generative model based on a discrete diffusion model specifically designed for images with discrete latent variables such as those found in vector-quantized (VQ)-VAE models. G2D2 overcomes the limitations of previous methods by using a continuous relaxation technique to optimize the parameters of a variational distribution, effectively bridging the gap between continuous and discrete domains.

Discrete diffusion models for image data often use a variant called "mask-absorbing" due to its efficiency. However, this model has a significant drawback in inverse-problem solving. While a substantial portion of the image structure is determined in the initial stages of generation in discrete diffusion models (i.e., when only a few tokens are determined), the mask-absorbing type does not allow transitions from an unmasked state to either a masked state or another unmasked state. Our experiments show that this restriction imposes a significant limitation on performance.

To address this issue, we use the star-shaped noise process previously proposed in the context of continuous diffusion models (Okhotin et al., 2024; Zhang et al., 2024). This process removes the dependency between consecutive sampling steps, thus expanding the solution space that can be explored. It therefore enables the correction of errors introduced in the early phases of sampling. Originally proposed to enhance the performance of diffusion models (Okhotin et al., 2024), it was later introduced as a decoupled noise annealing process in the context of inverse problems using continuous diffusion models, demonstrating its effectiveness (Zhang et al., 2024). In this study, we not only demonstrate that this process can be effectively applied to discrete diffusion models, but also find that it uniquely addresses potential issues inherent in mask-absorbing-type discrete diffusion processes, specifically the inability to correct errors introduced in the early stages of sampling during later steps.

We conduct an experimental investigation to evaluate the performance of G2D2 by comparing it to current methods using standard benchmark datasets. We consider methods that use both pixel-domain and latent diffusion models. We also explore the application of a discrete prior-based motion-data-generation model to solve an inverse problem, specifically path-conditioned generation, without requiring further training. The results of our study indicate that G2D2 shows promise in tackling various inverse problems by leveraging pre-trained discrete diffusion models.

## 2 PRELIMINARIES

### 2.1 DISCRETE DIFFUSION MODELS FOR IMAGE GENERATION

We first provide a brief overview of VQ-Diffusion (Gu et al., 2022; Tang et al., 2022), an image-generation model based on discrete diffusion processes. VQ-Diffusion generates images in a two-step process. It first produces discrete latent representations $\mathbf{z}_0$ using a discrete diffusion model trained on representations obtained from a pre-trained VQ-VAE model (Van Den Oord et al., 2017). It then transforms these representations into the continuous image space using a decoder. Each element of $\mathbf{z}_0 \in \{1, \ldots, K\}^{d_\mathbf{z}}$ corresponds to one of the embedding vectors from the codebook, denoted as $\mathbf{B} := \{\mathbf{b}_1, \ldots, \mathbf{b}_K\}, \mathbf{b}_k \in \mathbb{R}^{d_\mathbf{b}}$. During decoding, a variable $\mathbf{Z} \in \mathbf{B}^{d_\mathbf{z}}$ is constructed through codebook assignment, where $(\mathbf{Z})_i = \mathbf{b}_{z_{0,i}}$ and $z_{0,i}$ denotes the $i$-th element of $\mathbf{z}_0$. This variable is then fed into a continuous decoder $D : \mathbb{R}^{d_\mathbf{b} \times d_\mathbf{z}} \to \mathbb{R}^{d_{\mathbf{x}_0}}$ to obtain the final image: $\mathbf{x}_0 = D(\mathbf{Z})$.

In discrete diffusion models, a forward Markov process gradually corrupts the discrete latent representation $\mathbf{z}_0$, and a reverse process is learned to invert this process. A single step of the forward process of the Markov chain $\mathbf{z}_0 \to \cdots \to \mathbf{z}_t \to \cdots \to \mathbf{z}_T$ can be represented as,

$$q(z_{t,i}|\mathbf{z}_{t-1}) = \boldsymbol{v}^\mathsf{T}(z_{t,i})Q_t\boldsymbol{v}(z_{t-1,i}), \tag{1}$$

where $\boldsymbol{v}(z_{t,i})$ denotes a one-hot encoded vector representing the token at time step $t$, and $Q_t$ represents the transition matrix, which determines the probabilities of transitions between tokens. VQ-Diffusion uses a mask-absorbing-type forward process, which introduces a special masked token denoted as [MASK] in addition to the $K$ states from the VQ-VAE. The transition matrix is defined as

$$Q_t = \begin{pmatrix} \alpha_t + \beta_t & \beta_t & \beta_t & \cdots & 0 \\ \beta_t & \alpha_t + \beta_t & \beta_t & \cdots & 0 \\ \beta_t & \beta_t & \alpha_t + \beta_t & \cdots & 0 \\ \vdots & \vdots & \vdots & \ddots & \vdots \\ \gamma_t & \gamma_t & \gamma_t & \cdots & 1 \end{pmatrix}, \tag{2}$$

where the transition probabilities are determined by three parameters: $\alpha_t$, $\beta_t$, and $\gamma_t$. Specifically, $\alpha_t$ represents the probability of a token remaining unchanged, $\beta_t$ denotes the probability of transitioning to a different unmasked token, and $\gamma_t$ indicates the probability of the token being replaced with the [MASK] token. The probability $\beta_t$ between unmasked tokens is generally set to a very small value. These parameters are typically set so that $q(\mathbf{z}_T|\mathbf{z}_0)$ assigns all probability mass to the [MASK] token, and we also adopt this assumption.

During inference, the latent variable $\mathbf{z}_0$ corresponding to the clean image is obtained by executing the following reverse process:

$$p_\theta(\mathbf{z}_{t-1}|\mathbf{z}_t) = \sum_{\mathbf{z}_0} q(\mathbf{z}_{t-1}|\mathbf{z}_t, \mathbf{z}_0)\tilde{p}_\theta(\mathbf{z}_0|\mathbf{z}_t), \tag{3}$$

where $q(\mathbf{z}_{t-1}|\mathbf{z}_t, \mathbf{z}_0)$ represents the posterior distribution determined by the forward process, and $\tilde{p}_\theta$ denotes the denoising network that predicts the denoised token distribution at $t$. The output of $\tilde{p}_\theta$ is generally modeled as independent categorical distributions for each dimension in $\mathbf{z}_0$. In text-to-image models such as VQ-Diffusion, $\tilde{p}_\theta$ is trained with text conditioning. While the true distribution $q(\mathbf{z}_0)$ is not dimensionally independent, the whole Markov reverse process in (3) produces a distribution over categorical variables with correlations across dimensions. We distinguish between the clean distribution $\tilde{p}_\theta(\mathbf{z}_0|\mathbf{z}_t)$ estimated using the model and clean distribution $p_\theta(\mathbf{z}_0|\mathbf{z}_t)$ obtained through multiple reverse diffusion steps.

## 2.2 LINEAR-INVERSE-PROBLEM SETTINGS

Inverse problems involve estimating unknown data from measurement data. We specifically focus on linear inverse problems in the image domain. The relationship between the measurement image $\mathbf{y} \in \mathbb{R}^{d_{\mathbf{y}}}$ and unknown ground-truth image $\mathbf{x}_0 \in \mathbb{R}^{d_{\mathbf{x}_0}}$ can be represented as

$$\mathbf{y} = \mathbf{A}\mathbf{x}_0 + \boldsymbol{\eta}, \tag{4}$$

where $\mathbf{A} \in \mathbb{R}^{d_{\mathbf{y}} \times d_{\mathbf{x}_0}}$ is referred to as the forward linear operator, which describes the process by which the measurement data $\mathbf{y}$ are obtained from data $\mathbf{x}_0$. We assume this operator is known. The term $\boldsymbol{\eta}$ represents measurement noise, which we assume follows an isotropic Gaussian distribution with a known variance $\sigma_{\boldsymbol{\eta}}^2$. Consequently, the likelihood function $q(\mathbf{y}|\mathbf{x}_0)$ can be described as $\mathcal{N}(\mathbf{y}; \mathbf{A}\mathbf{x}_0, \sigma_{\boldsymbol{\eta}}^2\mathbf{I})$.

One of the primary challenges in inverse problems is their ill-posed nature. This means that for any given measurement $\mathbf{y}$, multiple candidate solutions may exist. To address this issue and determine $\mathbf{x}_0$, a common approach is to assume a prior distribution for $\mathbf{x}_0$, such as a Laplace distribution. Diffusion models have been utilized as more powerful and expressive priors, offering enhanced capabilities in solving these inverse problems (Kawar et al., 2022; Chung et al., 2023b; Wang et al., 2023; Rout et al., 2023). These methods are able to produce images that not only fit the measurement data but also exhibit high likelihood for the prior model. Given a prior $q(\mathbf{x}_0)$, the objective in the inverse problem is to sample from the posterior distribution $q(\mathbf{x}_0|\mathbf{y})$, which, according to Bayes' theorem, is proportional to $q(\mathbf{y}|\mathbf{x}_0)q(\mathbf{x}_0)$.

These methods can be categorized based on how they incorporate the information from the measurement data $\mathbf{y}$ into the generation trajectory of diffusion models. Methods such as denoising diffusion restoration models (DDRM) (Kawar et al., 2022) and denoising diffusion null-space models (DDNM) (Wang et al., 2023) leverage the assumption of linear operators, using singular value decomposition of the forward process to control the generative process. In contrast, methods such as diffusion posterior sampling (DPS) (Chung et al., 2023b) and posterior sampling with latent diffusion (PSLD) (Rout et al., 2023) operate by propagating the gradient of a loss term through the generative process. This loss term is designed to maximize the measurement likelihood, specifically by minimizing the term $\|\mathbf{y} - \mathbf{A}\mathbf{x}_0\|_2^2$.

However, the application of these methods to generative models that use discrete diffusion models as priors is not straightforward. This limitation stems from two primary factors. First, with the former methods, diffusion models are assumed trained in the pixel domain. Second, while the latter methods can be extended to latent diffusion-type models, they encounter difficulties when handling discrete diffusion models, in which the generative process is inherently discrete. The core challenge lies in the lack of a direct mechanism to propagate gradients of the loss function through the generative process in discrete diffusion models. In such models, after generating discrete data, a non-differentiable operation (i.e., codebook assignment) is followed by a decoding operation into continuous space, which prevents the application of conventional gradient-based guidance.

## 3 GRADIENT-GUIDED DISCRETE DIFFUSION, G2D2

Besides the lack of a straightforward mechanism to propagate gradients of the loss function through the generative process, a preliminary study reveals another main issue of directly applying the graphical model of a general mask-absorbing discrete diffusion model to sampling in an inverse-problem context. Figure 2 shows images decoded from $\mathbf{z}_0$, which are sampled from the denoising model $\tilde{p}_\theta(\mathbf{z}_0|\mathbf{z}_t)$ conditioned on the intermediate noisy discrete latent $\mathbf{z}_{90}$ or $\mathbf{z}_{80}$, along with the generated image from the full reverse process. These images are generated using a pre-trained VQ-Diffusion

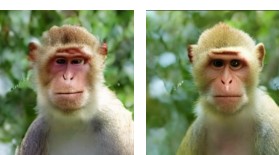

$t = 90$       $t = 80$       $t = 0$

Figure 2: At time step $t$, $\mathbf{z}_0$ is sampled from the prior model $\tilde{p}_\theta(\mathbf{z}_0|\mathbf{z}_t)$ and decoded. These results are generated with prompt "A face of monkey", without any guidance. **By initial ~10 steps ($t = 90$), coarse structure of image has already been determined.**

model (Gu et al., 2022) with the prompt "A face of monkey" (where $T = 100$). The results indicate

that the majority of the image structure is determined within the initial approximately 10% of the steps.

This observation highlights a problem for commonly used mask-absorbing discrete diffusion models in the context of inverse problems. The issue arises from the definition of $Q_t$ in (2), where $\beta_t$ is usually set to be extremely small. In the forward process, therefore, unmasked tokens are highly likely to either remain as identical unmasked tokens or transition to masked tokens. Masked tokens also remain unchanged thereafter.

This characteristic indicates that in the reverse process, the probability of unmasked tokens reverting to masked tokens or transitioning to different unmasked tokens is negligible. Consequently, when sampling to satisfy the measurement model, errors occurring in the early stages of sampling become nearly impossible to correct in the later phases.

One solution to this problem is the "re-masking" operation, which reverts unmasked tokens back to masked tokens. Similar approaches have been used with discrete predictor-corrector methods (Lezama et al., 2023) and predictor-corrector techniques for continuous-time discrete diffusion (Campbell et al., 2022; Zhao et al., 2024) to improve image-generation quality. However, those that involve reversing time steps can increase computational complexity. To address this, we demonstrate that by considering a noise process that is independent at each time step and different from the one used during the training of the prior, we can naturally resolve the inherent issues associated with mask-absorbing discrete diffusion models. We also show that despite this difference in the noise process, the prior model can still be used within our framework.

### 3.1 STAR-SHAPED NOISE PROCESS FOR INVERSE PROBLEM SOLVING

Inspired by Okhotin et al. (2024) and Zhang et al. (2024), G2D2 employs the star-shaped noise process. Figure 3 illustrates the differences between the Markov forward noise process (upper), which is used in general discrete diffusion models, and the star-shaped noise process (lower), both incorporating the relationship with the measurement $\mathbf{y}$. In the star-shaped noise process, the noisy discrete latents $\mathbf{z}_1, \ldots, \mathbf{z}_T$ are conditionally independent given $\mathbf{z}_0$. We assume that the distribution $q(\mathbf{z}_t|\mathbf{z}_0)$ adopts the same form as the original forward Markov process, specifically $q(z_{t,i}|\mathbf{z}_0) = \boldsymbol{v}^\mathsf{T}(z_{t,i})\overline{Q}_t\boldsymbol{v}(z_{0,i})$, with $\overline{Q}_t = Q_t \cdots Q_1$.

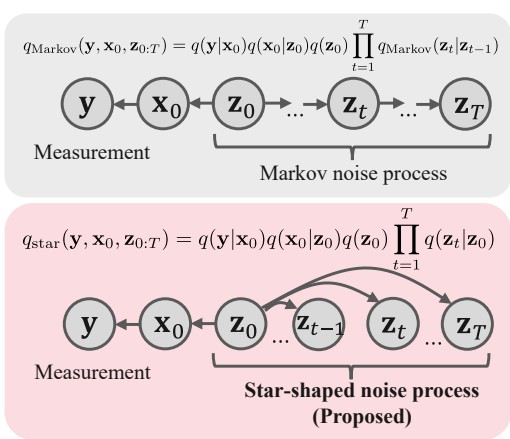

$$q_{\mathrm{Markov}}(\mathbf{y}, \mathbf{x}_0, \mathbf{z}_{0:T}) = q(\mathbf{y}|\mathbf{x}_0)q(\mathbf{x}_0|\mathbf{z}_0)q(\mathbf{z}_0)\prod_{t=1}^{T} q_{\mathrm{Markov}}(\mathbf{z}_t|\mathbf{z}_{t-1})$$

Measurement — Markov noise process

$$q_{\mathrm{star}}(\mathbf{y}, \mathbf{x}_0, \mathbf{z}_{0:T}) = q(\mathbf{y}|\mathbf{x}_0)q(\mathbf{x}_0|\mathbf{z}_0)q(\mathbf{z}_0)\prod_{t=1}^{T} q(\mathbf{z}_t|\mathbf{z}_0)$$

Measurement — **Star-shaped noise process (Proposed)**

Figure 3: Graphical models using Markov noise process (upper) and the star-shaped noise process (lower)

We aim to sample from the posterior $q_{\mathrm{star}}(\mathbf{z}_0|\mathbf{y})$ given the measurement $\mathbf{y}$ based on this graphical model. Given that the transformation from $\mathbf{z}_0$ to $\mathbf{x}_0$ is nearly deterministic in general decoders, we omit the random variable $\mathbf{x}_0$ in the subsequent discussion for simplicity.

To discuss the implementation of the sampling method based on the graphical model, we first introduce the conditional joint distribution $q_{\mathrm{sampling}}(\mathbf{z}_{0:T}|\mathbf{y}) = q(\mathbf{z}_T|\mathbf{y})\prod_{t=1}^{T} q(\mathbf{z}_{t-1}|\mathbf{z}_t, \mathbf{y})$. This conditional joint distribution has several properties:

**1. A single step of $q_{\mathrm{sampling}}$ inherently enables the "re-masking" operation.** In the star-shaped noise process, the positions of mask tokens in $\mathbf{z}_{t-1}$ and $\mathbf{z}_t$ are mutually independent and uncorrelated. Consequently, the conditional distribution $q(\mathbf{z}_{t-1}|\mathbf{z}_t, \mathbf{y})$ enables a "re-masking" operation, wherein unmasked tokens present in $\mathbf{z}_t$ can become masked tokens in $\mathbf{z}_{t-1}$. This property suggests that in mask-absorbing discrete diffusion, errors that occur in the initial stages of sampling can be corrected in subsequent steps, which provides an advantage when solving inverse problems.

**2. The marginal distribution $q_{\mathrm{sampling}}(\mathbf{z}_0|\mathbf{y})$ is identical to the target distribution $q_{\mathrm{star}}(\mathbf{z}_0|\mathbf{y})$ if the conditional distributions $q(\mathbf{z}_{t-1}|\mathbf{z}_t, \mathbf{y})$ are correctly specified based on the graphical model**

**of the star-shaped noise process.** The statement and proof are provided in the Appendix. This indicates that if sampling from each step $q(\mathbf{z}_{t-1}|\mathbf{z}_t, \mathbf{y})$ in $q_{\text{sampling}}$ is feasible, then it is possible to sample from the target marginal posterior $q_{\text{star}}(\mathbf{z}_0|\mathbf{y})$.

**3. The conditional joint distribution $q_{\text{sampling}}$ differs from that of the graphical model, i.e., $q_{\text{sampling}}(\mathbf{z}_{0:T}|\mathbf{y}) \neq q_{\text{star}}(\mathbf{z}_{0:T}|\mathbf{y})$.** In a standard Markov forward process, the decomposition of the joint distribution in a star-shaped noise process takes the form $q_{\text{star}}(\mathbf{z}_{0:T}|\mathbf{y}) = q(\mathbf{z}_T|\mathbf{y}) \prod_{t=1}^{T} q(\mathbf{z}_{t-1}|\mathbf{z}_{t:T}, \mathbf{y})$. However, $q_{\text{sampling}}$ deviates from this formulation by disregarding the dependencies on larger time steps, $\mathbf{z}_{t+1:T}$.

In subsequent sections, we introduce a variational distribution to approximate $q_{\text{sampling}}$, which inherently enables the re-masking operation based on Property 1. As established by Property 3, the joint distribution of $q_{\text{sampling}}$ differs from that of the star-shaped noise process graphical model. Nevertheless, Property 2 ensures that they share identical marginal distributions. Moreover, given that $q_{\text{sampling}}$ focuses on only two adjacent variables, we can formulate an algorithm to approximate its distribution using a variational approach.

## 3.2 G2D2 BASED ON STAR-SHAPED NOISE PROCESS

Based on the discussion in the previous section, we aim to implement $q_{\text{sampling}}$ on the graphical model of the star-shaped noise process, which inherently incorporates a re-masking process. Specifically, we introduce a variational distribution $p_{\boldsymbol{\alpha}}(\mathbf{z}_{0:T}|\mathbf{y})$ to approximate $q_{\text{sampling}}(\mathbf{z}_{0:T}|\mathbf{y})$, with the ultimate goal of ensuring that the marginal distribution $p_{\boldsymbol{\alpha}}(\mathbf{z}_0|\mathbf{y})$ approximates the true posterior $q(\mathbf{z}_0|\mathbf{y})$. The distribution $p_{\boldsymbol{\alpha}}$ is decomposed as

$$p_{\boldsymbol{\alpha}}(\mathbf{z}_{t-1}|\mathbf{z}_t, \mathbf{y}) = \sum_{\mathbf{z}_0} q(\mathbf{z}_{t-1}|\mathbf{z}_0)\tilde{p}_{\boldsymbol{\alpha}}(\mathbf{z}_0|\mathbf{z}_t, \mathbf{y}), \tag{5}$$

where $\tilde{p}_{\boldsymbol{\alpha}}(\mathbf{z}_0|\mathbf{z}_t, \mathbf{y})$ is a categorical distribution parameterized by $\boldsymbol{\alpha} \in \mathbb{R}^{T \times d_{\mathbf{z}} \times K}$, defined as $\tilde{p}_{\boldsymbol{\alpha}}(z_{0,i}|\mathbf{z}_t, \mathbf{y}) = \text{Cat}(z_{0,i}; \boldsymbol{\alpha}_{t,i,\cdot})$, i.e., $\tilde{p}_{\boldsymbol{\alpha}}(z_{0,i} = k|\mathbf{z}_t, \mathbf{y}) = \alpha_{t,i,k}$. This decomposition stems from the fact that the distribution $q(\mathbf{z}_{t-1}|\mathbf{z}_t, \mathbf{y})$ can be expressed as $\sum_{\mathbf{z}_0} q(\mathbf{z}_{t-1}|\mathbf{z}_0)q(\mathbf{z}_0|\mathbf{z}_t, \mathbf{y})$ based on the conditional independence. Note that both $q(\mathbf{z}_{t-1}|\mathbf{z}_0)$ and $\tilde{p}_{\boldsymbol{\alpha}}(\mathbf{z}_0|\mathbf{z}_t, \mathbf{y})$ have a mean field structure with independent categorical distributions across dimensions. Consequently, $p_{\boldsymbol{\alpha}}(\mathbf{z}_{t-1}|\mathbf{z}_t, \mathbf{y})$, obtained by marginalizing over $\mathbf{z}_0$, inherits this mean field property. For notational convenience, we denote the slice of distribution parameter $\boldsymbol{\alpha}$ at time step $t$ as $\boldsymbol{\alpha}_t \in \mathbb{R}^{d_{\mathbf{z}} \times K}$.

In G2D2, the variational distribution $p_{\boldsymbol{\alpha}}$ is obtained by optimizing an objective function derived from the following theorem:

**Theorem 3.1.** *Let $p_{\boldsymbol{\alpha}}$ be a distribution with the parameterization given by the decomposition in (5). Then, for any measurement $\mathbf{y}$, the following inequality holds for the Kullback-Leibler (KL) divergence between the marginal distributions:*

$$D_{\text{KL}}\left(p_{\boldsymbol{\alpha}}(\mathbf{z}_0|\mathbf{y}) \| q(\mathbf{z}_0|\mathbf{y})\right) \leq \sum_{t=1}^{T} \mathbb{E}_{\mathbf{z}_t \sim p_{\boldsymbol{\alpha}}(\mathbf{z}_t|\mathbf{y})}\left[D_{\text{KL}}\left(\tilde{p}_{\boldsymbol{\alpha}}(\mathbf{z}_0|\mathbf{z}_t, \mathbf{y}) \| q(\mathbf{z}_0|\mathbf{z}_t, \mathbf{y})\right)\right]. \tag{6}$$

*The full definitions of the terms are provided in the Appendix.*

The proof is provided in the Appendix. Based on this inequality, we aim to minimize each term in the sum on the right-hand side. Since $\tilde{p}_{\boldsymbol{\alpha}}$ is a different categorical distribution at each $t$, we minimize $\boldsymbol{\alpha}$ for each time step, ultimately aiming to minimize the left-hand side. Each term on the right-hand side of (6) takes the following form:

**Lemma 3.2.** *The KL divergence between the variational distribution $\tilde{p}_{\boldsymbol{\alpha}}(\mathbf{z}_0|\mathbf{z}_t, \mathbf{y})$ and true conditional $q(\mathbf{z}_0|\mathbf{z}_t, \mathbf{y})$ can be decomposed into two terms:*

$$D_{\text{KL}}\left(\tilde{p}_{\boldsymbol{\alpha}}(\mathbf{z}_0|\mathbf{z}_t, \mathbf{y}) \| q(\mathbf{z}_0|\mathbf{z}_t, \mathbf{y})\right) = D_{\text{KL}}\left(\tilde{p}_{\boldsymbol{\alpha}}(\mathbf{z}_0|\mathbf{z}_t, \mathbf{y}) \| q(\mathbf{z}_0|\mathbf{z}_t)\right) - \mathbb{E}_{\mathbf{z}_0 \sim \tilde{p}_{\boldsymbol{\alpha}}(\mathbf{z}_0|\mathbf{z}_t, \mathbf{y})}\left[\log q(\mathbf{y}|\mathbf{z}_0)\right], \tag{7}$$

*The full definitions of these terms are provided in the Appendix.*

This decomposition enables us to separately consider the fit to the prior and the consistency with the measurement data.

The first term on the right-hand side of (7) remains intractable. However, it is important to note that the star-shaped noise process shares the conditional distribution $q(\mathbf{z}_t|\mathbf{z}_0)$ with the original Markov noise process. Consequently, the reverse conditional distribution $q(\mathbf{z}_0|\mathbf{z}_t)$ will also be identical for both processes. Since the prior of the pre-trained discrete diffusion models is trained to approximate this distribution, we substitute this prior model $\tilde{p}_\theta(\mathbf{z}_0|\mathbf{z}_t)$ for $q(\mathbf{z}_0|\mathbf{z}_t)$ into the objective function of (7). This substitution transforms the term into a KL divergence between two categorical distributions, enabling the computation of gradients with respect to the parameter $\boldsymbol{\alpha}$.

The second term involves an expectation calculation over a categorical distribution, for which we use the Gumbel-Softmax re-parameterization trick (Jang et al., 2016; Maddison et al., 2016). The implementation of this trick is discussed in the subsequent section. This approach makes the term differentiable with respect to the categorical distribution's parameter $\boldsymbol{\alpha}$, facilitating continuous optimization. The explicit form of the resultant loss function is detailed in the Appendix.

Based on Theorem 3.1 and Lemma 3.2, G2D2 optimizes the parameter $\boldsymbol{\alpha}$ of $p_{\boldsymbol{\alpha}}$ for $t = T, \ldots, 1$ while sequentially sampling $\mathbf{z}_{0:T}$. In the optimization step, any continuous optimization method, such as Adam (Kingma, 2014), can be used. Implementation considerations are discussed in the following section. This algorithm is detailed in Algorithm 1, and G2D2 is illustrated in Figure 1.

---

**Algorithm 1** Gradient-Guided Discrete Diffusion, **G2D2**

---

**Require:** Input condition $\mathbf{y}$, pre-trained discrete diffusion model $p_\theta$, forget coefficient $\gamma$
1: $\mathbf{z}_T \sim q(\mathbf{z}_T)$
2: **for** $t = T, \ldots, 1$ **do**
3:     **if** $t = T$ **then**
4:         Initialize: $\boldsymbol{\alpha}_t = \log \tilde{p}_\theta(\mathbf{z}_0|\mathbf{z}_t)$
5:     **else**
6:         Initialize: $\boldsymbol{\alpha}_t = \exp(\gamma \log \boldsymbol{\alpha}_{t+1} + (1 - \gamma) \log \tilde{p}_\theta(\mathbf{z}_0|\mathbf{z}_t))$
7:     **end if**
8:     // continuous optimization
9:     $\boldsymbol{\alpha}_t = \arg\min_{\boldsymbol{\alpha}_t} D_{\mathrm{KL}}\left(\tilde{p}_{\boldsymbol{\alpha}}(\mathbf{z}_0|\mathbf{z}_t, \mathbf{y}) \| \tilde{p}_\theta(\mathbf{z}_0|\mathbf{z}_t)\right) - \mathbb{E}_{\mathbf{z}_0 \sim \tilde{p}_{\boldsymbol{\alpha}}(\mathbf{z}_0|\mathbf{z}_t, \mathbf{y})} \left[\log q(\mathbf{y}|\mathbf{z}_0)\right]$
10:     Sample $\mathbf{z}_{t-1} \sim p_{\boldsymbol{\alpha}}(\mathbf{z}_{t-1}|\mathbf{z}_t, \mathbf{y}) = \sum_{\mathbf{z}_0} q(\mathbf{z}_{t-1}|\mathbf{z}_0)\tilde{p}_{\boldsymbol{\alpha}}(\mathbf{z}_0|\mathbf{z}_t, \mathbf{y})$
11: **end for**
12: **return** $\mathbf{x}_0$ by decoding $\mathbf{z}_0$

---

### 3.3 IMPLEMENTATION CONSIDERATIONS

**Gumbel-Softmax dequantization**    We use the Gumbel-Softmax trick (Jang et al., 2016; Maddison et al., 2016) to make the computation of the second term in (7) differentiable. At time step $t$, this process begins by generating Gumbel-Softmax samples using parameters of $\tilde{p}_{\boldsymbol{\alpha}}$ as follows: $\hat{z}_{0,i,k} = \mathrm{softmax}\left((\log \alpha_{t,i,k} + g_{i,k})/\tau\right)$, where $g_{i,k}$ represents samples drawn from the Gumbel distribution, and $\tau$ is the temperature parameter. This procedure generates a "soft" categorical sample for each dimension in $\mathbf{z}_0$, indicating the proportional selection of each codebook element. As these proportions correspond to the contribution rate of each codebook element, we construct $\mathbf{Z}_{\mathrm{Gumbel}} \in \mathbb{R}^{d_{\mathbf{z}} \times d_b}$ as their weighted sum: $(\mathbf{Z}_{\mathrm{Gumbel}})_i = \sum_{k=1}^{K} \hat{z}_{0,i,k} \mathbf{b}_k$. Finally, we pass $\mathbf{Z}_{\mathrm{Gumbel}}$ through the decoder to obtain the image $\mathbf{x}_0 = D(\mathbf{Z}_{\mathrm{Gumbel}})$. By substituting this image into the likelihood function $q(\mathbf{y}|\mathbf{x}_0)$, we have the differentiable objective with respect to the variational parameter $\boldsymbol{\alpha}_t$, enabling continuous optimization. For linear inverse problems, the objective function will include the term $\|\mathbf{y} - \mathbf{A}\mathbf{x}_0(\boldsymbol{\alpha}_t)\|_2^2$, excluding the constant term derived from measurement noise.

**Optimization initialization strategy**    At time step $t$, we are required to optimize the variational parameter $\boldsymbol{\alpha}_t$. To expedite this process, we can leverage the optimized values from the previous time step as the initialization for the optimization process, effectively reducing the number of required optimization steps. To achieve this, we introduce a forgetting coefficient $\gamma$ and initialize $\boldsymbol{\alpha}_t$ through a weighted sum of the previous optimized variables and the prior model's output in the logarithm domain, given by $\boldsymbol{\alpha}_t = \exp(\gamma \log \boldsymbol{\alpha}_{t+1} + (1 - \gamma) \log \tilde{p}_\theta(\mathbf{z}_0|\mathbf{z}_t))$.

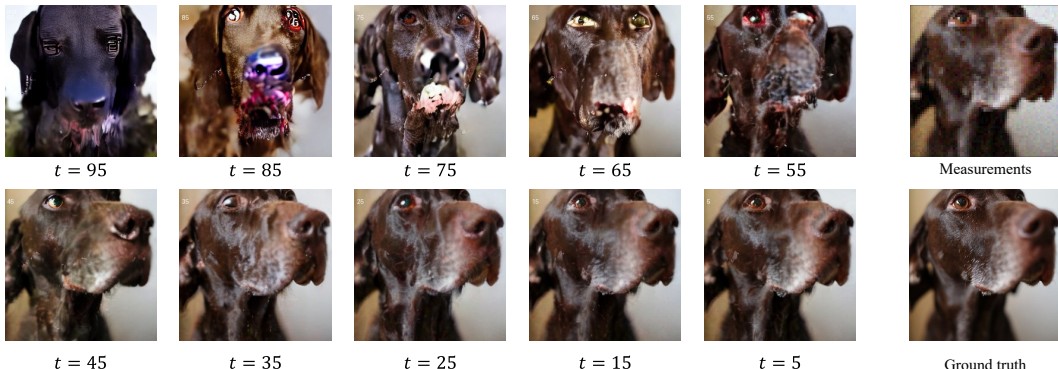

Figure 4: Images sampled from the prior model $\tilde{p}_\theta(\mathbf{z}_0|\mathbf{z}_t)$ using intermediate $\mathbf{z}_t$ during the process of G2D2 in image inverse problem solving. The progression demonstrates how initial structural errors are gradually corrected as the sampling proceeds in G2D2.

### 3.4 APPLICATION OF G2D2 TO MASKED GENERATIVE MODELS

As discussed in (Zheng et al., 2024), mask-absorbing discrete diffusion models and masked generative models, such as MaskGIT (Chang et al., 2022), share a similar framework. Except for temporal conditioning, these models are nearly identical and are trained to approximate $q(\mathbf{z}_0|\mathbf{z}_t)$. Therefore, G2D2 can be straightforwardly applied to masked generative models. We give an example of solving inverse problems using a masked generative model as a prior model for motion data in the following section.

## 4 EXPERIMENTS

### 4.1 EXPERIMENTAL SETUP

We evaluate G2D2 on inverse problems in image processing and compare it with other diffusion model-based inverse-problem-solving methods. We also demonstrate gradient-based guidance on a discrete-latent variable-based motion-domain generative model without additional training, showing the applicability of G2D2 to other domains.

**Image inverse problems and evaluation metrics** We conduct experiments on two tasks: (1) super-resolution (SR) and (2) Gaussian deblurring. For the SR task, the linear forward operator downscales the image by a factor of 4 using a bicubic resizer. For the Gaussian-deblurring task, we set the kernel size to $61 \times 61$ with a Gaussian kernel standard deviation of 3.0. The measurements are obtained by applying the forward operator to the ground truth images normalized to the range $[-1, 1]$, followed by the addition of Gaussian noise with a standard deviation of 0.05. As metrics, we use the learned perceptual image patch similarity (LPIPS) (Zhang et al., 2018) score to measure perceptual proximity to the original image, and the peak signal-to-noise ratio (PSNR) to measure the closeness of the signal.

**Datasets** Following previous studies, we use the ImageNet (Deng et al., 2009) and Flickr-Faces-HQ (FFHQ) (Karras et al., 2019) datasets. The size of both datasets is 256×256. For comparison, we use a subset of 100 images from each validation set.

**Baselines** We compare DPS (Chung et al., 2023b), DDRM (Kawar et al., 2022), which use diffusion models trained in the pixel domain, and PSLD (Rout et al., 2023) and ReSample (Song et al., 2024), which use diffusion models trained in the latent space acquired from VAE (latent diffusion models) as baselines with G2D2.

**Implementation details**     Regarding G2D2, for both the ImageNet and FFHQ experiments, we use a pre-trained VQ-Diffusion model [1] that is trained on the ITHQ dataset (Tang et al., 2022). In all experiments, we optimize the parameters $\alpha_t$ of the variational categorical distribution within the G2D2 algorithm's optimization step using the Adam optimizer Kingma (2014). To balance the prior and likelihood terms in the objective function, we introduce hyperparameters. For the image inverse problem experiments, we used text prompts for the VQ-Diffusion model: "`a photo of [Class Name]`" for ImageNet and "`a high-quality headshot of a person`" for FFHQ.

Details of the experiments and comparison methods are provided in the Appendix.

## 4.2   IMAGE INVERSE PROBLEM SOLVING ON IMAGENET AND FFHQ

Figure 5 shows the qualitative results of image inverse problem solving, and Tables 1 and 2 list the quantitative results. G2D2 performs comparably to the other methods using diffusion models trained in the continuous domain. Note that the pre-trained models used for each method are different, which particularly contributes to the superiority of pixel-domain methods on FFHQ. With DDRM, it is assumed that the amount of measurement noise is known and require the singular value decomposition of the linear operator. We also show images in the intermediate phase of the G2D2 algorithm in Figure 4.

Table 1: Quantitative evaluation on ImageNet 256×256. Performance comparison of different methods on various linear tasks in image domain. Values show the mean over 100 images.

| Prior Type | Method | SR (×4) | | Gaussian deblurring | |
|---|---|---|---|---|---|
| | | LPIPS↓ | PSNR↑ | LPIPS↓ | PSNR↑ |
| Pixel-domain | DPS (Chung et al., 2023b) | 0.367 | 22.61 | 0.443 | 19.04 |
| | DDRM (Kawar et al., 2022) | 0.352 | 24.00 | 0.246 | 27.30 |
| LDM | PSLD (Rout et al., 2023) | 0.332 | 24.43 | 0.365 | 24.04 |
| | ReSample (Song et al., 2024) | 0.382 | 22.63 | 0.438 | 22.32 |
| Discrete | G2D2 (proposed) | 0.349 | 23.20 | 0.375 | 22.71 |
| | G2D2 w/ Markov noise process | 0.409 | 21.48 | 0.431 | 21.78 |

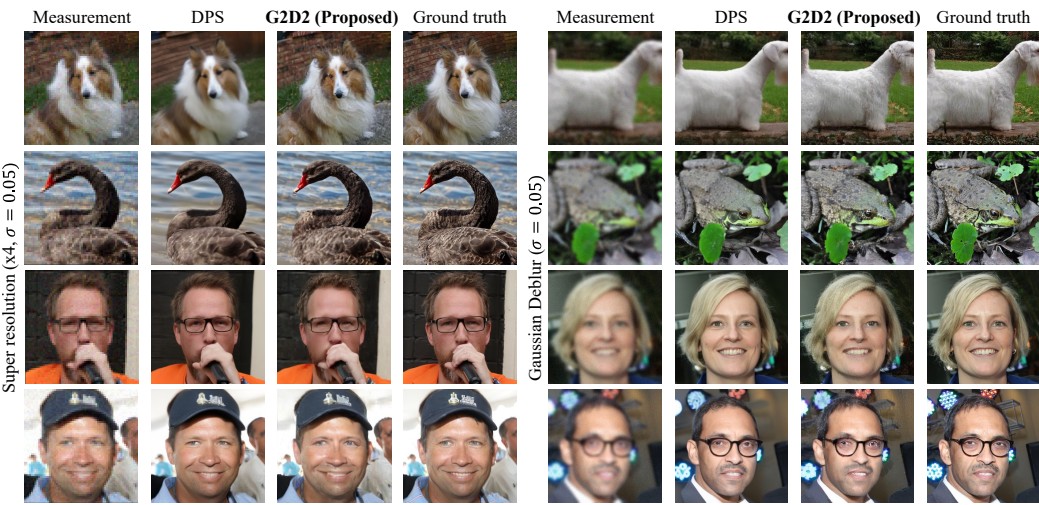

Figure 5: Qualitative results of G2D2 and DPS.

## 4.3   ABLATION STUDY ON GRAPHICAL MODELS

It is possible to derive a similar algorithm to G2D2 that uses a Markov noise process as the graphical model. However, as discussed at the beginning of Section 3, this graphical model does not allow

---

[1] `https://huggingface.co/microsoft/vq-diffusion-ithq`

Table 2: Quantitative evaluation on FFHQ 256×256. Performance comparison of different methods on various linear tasks in image domain. Values show mean over 100 images.

| Prior Type | Method | SR (×4) | | Gaussian deblurring | |
|---|---|---|---|---|---|
| | | LPIPS↓ | PSNR↑ | LPIPS↓ | PSNR↑ |
| Pixel-domain | DPS (Chung et al., 2023b) | 0.227 | 26.73 | 0.225 | 26.02 |
| | DDRM (Kawar et al., 2022) | 0.242 | 28.23 | 0.201 | 31.12 |
| LDM | PSLD (Rout et al., 2023) | 0.276 | 27.62 | 0.304 | 27.37 |
| | ReSample (Song et al., 2024) | 0.507 | 22.98 | 0.329 | 25.69 |
| Discrete | G2D2 (proposed) | 0.271 | 26.93 | 0.287 | 26.35 |
| | G2D2 w/ Markov noise process | 0.395 | 23.94 | 0.365 | 25.16 |

for the "re-masking" operation, which means it cannot correct errors that occur early in the sampling process. We refer to this variant as **G2D2 w/ Markov noise process**, and its performance is presented in Tables 1 and 2 on ImageNet and FFHQ, respectively. Additional qualitative results are provided in the Appendix C.3. The results indicate that the introduction of the star-shaped noise process significantly improves performance, making G2D2 comparable to continuous-based methods.

### 4.4 MOTION INVERSE PROBLEM SOLVING

As discussed in Section 3.4, our method can also be applied to Masked generative models. We conduct experiments to manipulate Generative Masked Motion Model (MMM) Pinyoanuntapong et al. (2024), a generative model for motion data, using gradient guidance. Specifically, we perform a **path following task** where generation is conditioned on the position information of the hip joint. Since joint position information can be calculated from motion data, this can also be treated within the framework of inverse problems. While there have been examples of achieving path following in motion generation models with continuous latent spaces (Song et al., 2023b; Uchida et al., 2024), we are the first to accomplish this using a motion generation model with discrete latent variables in a training-free manner. Appendix C.11 provides additional samples and detailed experimental information.

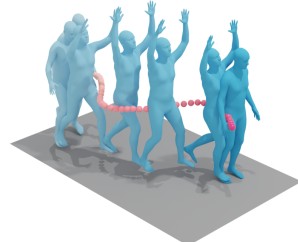

Figure 6: Results of executing G2D2 using a motion generation model as a prior in the motion inverse problem (path following generation task).

## 5 CONCLUSION

We proposed G2D2 for solving inverse problems using discrete diffusion models as priors. We demonstrated that G2D2 effectively addresses the limitation of discrete diffusion in inverse problem-solving by using a continuous relaxation technique and star-shaped noise process. Specifically, G2D2 approximates the posterior in inverse problems by optimizing the parameters of a variational distribution, composed of parameterized categorical distributions, at each time step of the diffusion process. Our experiments show that G2D2 performs comparable to its continuous counterparts, opening up possibilities for training-free applications of discrete diffusion models across a wide range of tasks.

**Limitations and future works** G2D2 does not significantly surpass its continuous counterparts in terms of computational speed or performance. We anticipate that these limitations can be mitigated through the optimization of efficiency and the enhancement of prior models. The application to more complex problem settings, including nonlinear inverse problems, as well as to other domains such as audio and video, constitutes future work.

**Ethics statement** Our G2D2 method, which uses discrete diffusion models as priors for solving inverse problems, carries potential risks similar to those of previously proposed techniques in this field. We acknowledge that these methods, including ours, may inadvertently perpetuate biases present in training data or be misused for generating misleading or harmful content. We are committed to addressing these ethical concerns and promoting responsible use of our technology. We urge users of our method to exercise caution and consider the ethical implications of its applications.

**Reproducibility statement** We will provide as detailed a description as possible regarding the reproduction of experiments in the Appendix, and we plan to release our code when this paper is published.

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

## A  RELATED WORK

In this section, we review the relevant prior works.

## A.1 LEVERAGING DIFFUSION MODELS AS PRIOR MODELS FOR INVERSE PROBLEMS

**Pixel-Domain Diffusion Models for Inverse Problems** Several methods have been proposed that utilize pixel-domain diffusion models for solving inverse problems. DDRM and DDNM (Kawar et al., 2022; Wang et al., 2023) assume linear operators and known noise levels, leveraging the singular value decomposition (SVD) of these operators. ΠGDM (Song et al., 2023a) can handle certain classes of non-linear operators, such as low dynamic range, where a pseudo-inverse operator can be defined. Notably, ΠGDM does not require SVD or gradient computations for such a case.

DPS (Chung et al., 2023b) broadens the applicability to cases where operator gradients can be computed, enabling it to handle both linear and non-linear operators like phase retrieval and non-linear blur. Other notable methods in this category include RePaint (Lugmayr et al., 2022) and RED-Diff (Mardani et al., 2024).

**Latent Diffusion Models for Inverse Problems** Recent work has also explored the use of latent diffusion models for inverse problems. PSLD (Rout et al., 2023) extends the ideas of DPS to latent diffusion models, demonstrating provable sample recovery for linear inverse problems. ReSample (Song et al., 2024) achieves data consistency by solving an optimization problem at each step during sampling.

Of particular relevance to our work is DAPS (Zhang et al., 2024), which, like our approach, adopts a graphical model during sampling that differs from the one used during training of the prior model. This approach, known as the noise decoupling scheme, offers new possibilities for adapting diffusion models to various inverse problems.

**Application of Inverse Problem Solving in Various Domains** Solving inverse problems using diffusion models has enabled various real-world applications. In the image domain, diffusion models have been extensively studied and applied to tasks such as image deblurring, super-resolution, and inpainting (Lugmayr et al., 2022; Chung et al., 2023a; Zhu et al., 2023). In the audio domain, methods such as those proposed by Song et al. (2021a), Chung et al. (2023c), and Bian et al. (2024) have been developed to address tasks like dereverberation and audio restoration. Similarly, in the medical imaging domain, approaches like those introduced by Song et al. (2021a), Chung et al. (2023c), and Bian et al. (2024) have been used to improve image reconstruction and enhance diagnostic accuracy. These advancements demonstrate the versatility and effectiveness of diffusion models across different domains.

## A.2 CONDITIONAL GENERATION USING DISCRETE DIFFUSION MODELS AS PRIORS

While our work focuses on inverse problems, it is important to consider related approaches in conditional generation tasks using discrete diffusion models as priors. These methods, primarily developed in the context of graph generation and protein design, introduce new conditioning to pre-trained models rather than directly addressing inverse problems.

The predominant strategy in this field involves training additional guidance networks. For instance, in protein sequence generation, LaMBO-2 (Gruver et al., 2024) and Cemri et al. (2024) learn networks that evaluate how intermediate features of samples during generation achieve the desired objectives. Similarly, CGD (Klarner et al., 2024) learns a guidance model for corrupted data. Other examples requiring additional training include Nisonoff et al. (2024) and DiGress (Vignac et al., 2023) for graph generation.

In contrast, Chen et al. (2024b) proposes a training-free approach to guide discrete diffusion models for generating Electronic Health Record data. This method employs Langevin dynamics sampling to minimize a given loss function by adjusting the parameters of the final layer of the prior model's transformer output. However, this approach faces scalability issues with models having large discrete latent spaces, such as VQ-Diffusion, as it requires evaluating all possible discrete states to compute the loss function.

Another training-free method for guiding generative models with discrete latents is proposed by Li et al. (2024). This approach avoids gradient computation of the loss function, instead evaluating the loss on multiple generated samples and conducting sampling based on these values. However, like

Chen et al. (2024b), this method is expected to be inefficient for models with relatively large discrete latent spaces.

## B  PROOFS

**Full Statement of Theorem 3.1**  In this section, we provide detailed proofs for the main theoretical results presented in the paper.

**Theorem B.1** (Full version of Theorem 3.1). *Let $p_{\boldsymbol{\alpha}}$ be a distribution with the parameterization given by the decomposition in (5). Then, for any measurements $\mathbf{y}$, the following inequality holds for the KL divergence between the marginal distributions:*

$$D_{\mathrm{KL}}\left(p_{\boldsymbol{\alpha}}(\mathbf{z}_0|\mathbf{y})\|q(\mathbf{z}_0|\mathbf{y})\right) \leq \sum_{t=1}^{T} \mathbb{E}_{\mathbf{z}_t \sim p_{\boldsymbol{\alpha}}(\mathbf{z}_t|\mathbf{y})}\left[D_{\mathrm{KL}}\left(\tilde{p}_{\boldsymbol{\alpha}}(\mathbf{z}_0|\mathbf{z}_t,\mathbf{y})\|q(\mathbf{z}_0|\mathbf{z}_t,\mathbf{y})\right)\right], \quad (8)$$

*where $p_{\boldsymbol{\alpha}}(\mathbf{z}_0|\mathbf{y})$ is the variational marginal distribution parameterized by $\boldsymbol{\alpha}$, $q(\mathbf{z}_0|\mathbf{y})$ is the true posterior distribution, $\tilde{p}_{\boldsymbol{\alpha}}(\mathbf{z}_0|\mathbf{z}_t,\mathbf{y})$ is the variational conditional distribution as defined in (5), $q(\mathbf{z}_0|\mathbf{z}_t,\mathbf{y})$ is the true conditional distribution, $p_{\boldsymbol{\alpha}}(\mathbf{z}_t|\mathbf{y})$ is the marginal distribution at time step $t$, and $T$ is the total number of time steps in the diffusion process.*

*Proof.* To prove the inequality in Theorem 3.1, we start by noting that the KL divergence between the marginal distributions $p_{\boldsymbol{\alpha}}(\mathbf{z}_0|\mathbf{y})$ and $q(\mathbf{z}_0|\mathbf{y})$ can be bounded by the KL divergence between the joint distributions $p_{\boldsymbol{\alpha}}(\mathbf{z}_{0:T}|\mathbf{y})$ and $q(\mathbf{z}_{0:T}|\mathbf{y})$:

$$D_{\mathrm{KL}}\left(p_{\boldsymbol{\alpha}}(\mathbf{z}_0|\mathbf{y}) \| q(\mathbf{z}_0|\mathbf{y})\right) \leq D_{\mathrm{KL}}\left(p_{\boldsymbol{\alpha}}(\mathbf{z}_{0:T}|\mathbf{y}) \| q(\mathbf{z}_{0:T}|\mathbf{y})\right). \quad (9)$$

This inequality holds because marginalization cannot increase the KL divergence between distributions.

Next, we decompose the joint KL divergence using the chain rule and the definitions of the distributions:

$$
\begin{aligned}
D_{\mathrm{KL}}\left(p_{\boldsymbol{\alpha}}(\mathbf{z}_{0:T}|\mathbf{y}) \| q(\mathbf{z}_{0:T}|\mathbf{y})\right) &= \mathbb{E}_{p_{\boldsymbol{\alpha}}(\mathbf{z}_{0:T}|\mathbf{y})}\left[\log \frac{p_{\boldsymbol{\alpha}}(\mathbf{z}_{0:T}|\mathbf{y})}{q(\mathbf{z}_{0:T}|\mathbf{y})}\right] \\
&= \mathbb{E}_{p_{\boldsymbol{\alpha}}(\mathbf{z}_{0:T}|\mathbf{y})}\left[\log \frac{p_{\boldsymbol{\alpha}}(\mathbf{z}_T|\mathbf{y})\prod_{t=1}^{T} p_{\boldsymbol{\alpha}}(\mathbf{z}_{t-1}|\mathbf{z}_t,\mathbf{y})}{q(\mathbf{z}_T|\mathbf{y})\prod_{t=1}^{T} q(\mathbf{z}_{t-1}|\mathbf{z}_t,\mathbf{y})}\right] \\
&= \mathbb{E}_{p_{\boldsymbol{\alpha}}(\mathbf{z}_{0:T}|\mathbf{y})}\left[\log \frac{p_{\boldsymbol{\alpha}}(\mathbf{z}_T|\mathbf{y})}{q(\mathbf{z}_T|\mathbf{y})} + \sum_{t=1}^{T}\log \frac{p_{\boldsymbol{\alpha}}(\mathbf{z}_{t-1}|\mathbf{z}_t,\mathbf{y})}{q(\mathbf{z}_{t-1}|\mathbf{z}_t,\mathbf{y})}\right] \\
&= D_{\mathrm{KL}}\left(p_{\boldsymbol{\alpha}}(\mathbf{z}_T|\mathbf{y}) \| q(\mathbf{z}_T|\mathbf{y})\right) + \sum_{t=1}^{T}\mathbb{E}_{p_{\boldsymbol{\alpha}}(\mathbf{z}_{0:T}|\mathbf{y})}\left[\log \frac{p_{\boldsymbol{\alpha}}(\mathbf{z}_{t-1}|\mathbf{z}_t,\mathbf{y})}{q(\mathbf{z}_{t-1}|\mathbf{z}_t,\mathbf{y})}\right].
\end{aligned}
$$
$$(10)$$

In the context of mask-absorbing state diffusion, the distribution $p_{\boldsymbol{\alpha}}(\mathbf{z}_T|\mathbf{y})$ is the same as $q(\mathbf{z}_T|\mathbf{y})$ because $\mathbf{z}_T$ is fully determined by the diffusion process and is independent of $\boldsymbol{\alpha}$. Therefore, the first term is zero:

$$D_{\mathrm{KL}}\left(p_{\boldsymbol{\alpha}}(\mathbf{z}_T|\mathbf{y}) \| q(\mathbf{z}_T|\mathbf{y})\right) = 0. \quad (11)$$

This simplifies (10) to:

$$D_{\mathrm{KL}}\left(p_{\boldsymbol{\alpha}}(\mathbf{z}_{0:T}|\mathbf{y}) \| q(\mathbf{z}_{0:T}|\mathbf{y})\right) = \sum_{t=1}^{T}\mathbb{E}_{p_{\boldsymbol{\alpha}}(\mathbf{z}_{0:T}|\mathbf{y})}\left[\log \frac{p_{\boldsymbol{\alpha}}(\mathbf{z}_{t-1}|\mathbf{z}_t,\mathbf{y})}{q(\mathbf{z}_{t-1}|\mathbf{z}_t,\mathbf{y})}\right]. \quad (12)$$

We can further simplify the expectation over $\mathbf{z}_{0:T}$ by focusing on $\mathbf{z}_t$ and $\mathbf{z}_{t-1}$:

$$D_{\mathrm{KL}}\left(p_{\boldsymbol{\alpha}}(\mathbf{z}_{0:T}|\mathbf{y}) \,\|\, q(\mathbf{z}_{0:T}|\mathbf{y})\right) = \sum_{t=1}^{T} \mathbb{E}_{\mathbf{z}_t \sim p_{\boldsymbol{\alpha}}(\mathbf{z}_t|\mathbf{y})} \left[D_{\mathrm{KL}}\left(p_{\boldsymbol{\alpha}}(\mathbf{z}_{t-1}|\mathbf{z}_t,\mathbf{y}) \,\|\, q(\mathbf{z}_{t-1}|\mathbf{z}_t,\mathbf{y})\right)\right]. \quad (13)$$

Now, for each term in the sum, we apply the chain rule for KL divergence to relate $\mathbf{z}_{t-1}$ and $\mathbf{z}_0$:

$$\begin{aligned}
&D_{\mathrm{KL}}\left(p_{\boldsymbol{\alpha}}(\mathbf{z}_{t-1}|\mathbf{z}_t,\mathbf{y}) \,\|\, q(\mathbf{z}_{t-1}|\mathbf{z}_t,\mathbf{y})\right) \\
&\quad + \mathbb{E}_{\mathbf{z}_{t-1} \sim p_{\boldsymbol{\alpha}}(\mathbf{z}_{t-1}|\mathbf{z}_t,\mathbf{y})} \left[D_{\mathrm{KL}}\left(\tilde{p}_{\boldsymbol{\alpha}}(\mathbf{z}_0|\mathbf{z}_{t-1},\mathbf{z}_t,\mathbf{y}) \,\|\, q(\mathbf{z}_0|\mathbf{z}_{t-1},\mathbf{z}_t,\mathbf{y})\right)\right] \\
&= D_{\mathrm{KL}}\left(\tilde{p}_{\boldsymbol{\alpha}}(\mathbf{z}_0|\mathbf{z}_t,\mathbf{y}) \,\|\, q(\mathbf{z}_0|\mathbf{z}_t,\mathbf{y})\right) \\
&\quad + \mathbb{E}_{\mathbf{z}_0 \sim \tilde{p}_{\boldsymbol{\alpha}}(\mathbf{z}_0|\mathbf{z}_t,\mathbf{y})} \left[D_{\mathrm{KL}}\left(p_{\boldsymbol{\alpha}}(\mathbf{z}_{t-1}|\mathbf{z}_0,\mathbf{z}_t,\mathbf{y}) \,\|\, q(\mathbf{z}_{t-1}|\mathbf{z}_0,\mathbf{z}_t,\mathbf{y})\right)\right]. \quad (14)
\end{aligned}$$

In this equation, the left-hand side represents the KL divergence between $p_{\boldsymbol{\alpha}}$ and $q$ at time $t-1$ conditioned on $\mathbf{z}_t$, plus the expected KL divergence between their respective conditional distributions of $\mathbf{z}_0$. The right-hand side represents the KL divergence between $\tilde{p}_{\boldsymbol{\alpha}}$ and $q$ directly conditioned on $\mathbf{z}_t$, plus an expected KL divergence over $\mathbf{z}_0$.

The crucial observation here is that the last term on the right-hand side is zero. This is because $p_{\boldsymbol{\alpha}}$ and $q$ share the same reverse diffusion process when conditioned on $\mathbf{z}_0$ and $\mathbf{z}_t$, i.e.,

$$\begin{aligned}
p_{\boldsymbol{\alpha}}(\mathbf{z}_{t-1}|\mathbf{z}_0,\mathbf{z}_t,\mathbf{y}) &= q(\mathbf{z}_{t-1}|\mathbf{z}_0) \\
&= q(\mathbf{z}_{t-1}|\mathbf{z}_0,\mathbf{z}_t,\mathbf{y}). \quad (15)
\end{aligned}$$

Therefore, the KL divergence between these conditional distributions is zero:

$$D_{\mathrm{KL}}\left(p_{\boldsymbol{\alpha}}(\mathbf{z}_{t-1}|\mathbf{z}_0,\mathbf{z}_t,\mathbf{y}) \,\|\, q(\mathbf{z}_{t-1}|\mathbf{z}_0,\mathbf{z}_t,\mathbf{y})\right) = 0. \quad (16)$$

Substituting back into (14), we obtain:

$$\begin{aligned}
&D_{\mathrm{KL}}\left(p_{\boldsymbol{\alpha}}(\mathbf{z}_{t-1}|\mathbf{z}_t,\mathbf{y}) \,\|\, q(\mathbf{z}_{t-1}|\mathbf{z}_t,\mathbf{y})\right) \\
&= D_{\mathrm{KL}}\left(\tilde{p}_{\boldsymbol{\alpha}}(\mathbf{z}_0|\mathbf{z}_t,\mathbf{y}) \,\|\, q(\mathbf{z}_0|\mathbf{z}_t,\mathbf{y})\right) \\
&\quad - \mathbb{E}_{\mathbf{z}_{t-1} \sim p_{\boldsymbol{\alpha}}(\mathbf{z}_{t-1}|\mathbf{z}_t,\mathbf{y})} \left[D_{\mathrm{KL}}\left(\tilde{p}_{\boldsymbol{\alpha}}(\mathbf{z}_0|\mathbf{z}_{t-1},\mathbf{z}_t,\mathbf{y}) \,\|\, q(\mathbf{z}_0|\mathbf{z}_{t-1},\mathbf{z}_t,\mathbf{y})\right)\right]. \quad (17)
\end{aligned}$$

Since the KL divergence is always non-negative, the expected KL divergence on the right-hand side is non-negative, which implies:

$$D_{\mathrm{KL}}\left(p_{\boldsymbol{\alpha}}(\mathbf{z}_{t-1}|\mathbf{z}_t,\mathbf{y}) \,\|\, q(\mathbf{z}_{t-1}|\mathbf{z}_t,\mathbf{y})\right) \leq D_{\mathrm{KL}}\left(\tilde{p}_{\boldsymbol{\alpha}}(\mathbf{z}_0|\mathbf{z}_t,\mathbf{y}) \,\|\, q(\mathbf{z}_0|\mathbf{z}_t,\mathbf{y})\right). \quad (18)$$

Substituting (18) back into (13), we obtain an upper bound on the joint KL divergence:

$$D_{\mathrm{KL}}\left(p_{\boldsymbol{\alpha}}(\mathbf{z}_{0:T}|\mathbf{y}) \,\|\, q(\mathbf{z}_{0:T}|\mathbf{y})\right) \leq \sum_{t=1}^{T} \mathbb{E}_{\mathbf{z}_t \sim p_{\boldsymbol{\alpha}}(\mathbf{z}_t|\mathbf{y})} \left[D_{\mathrm{KL}}\left(\tilde{p}_{\boldsymbol{\alpha}}(\mathbf{z}_0|\mathbf{z}_t,\mathbf{y}) \,\|\, q(\mathbf{z}_0|\mathbf{z}_t,\mathbf{y})\right)\right]. \quad (19)$$

Combining (9) and (19), we conclude:

$$D_{\mathrm{KL}}\left(p_{\boldsymbol{\alpha}}(\mathbf{z}_0|\mathbf{y}) \,\|\, q(\mathbf{z}_0|\mathbf{y})\right) \leq \sum_{t=1}^{T} \mathbb{E}_{\mathbf{z}_t \sim p_{\boldsymbol{\alpha}}(\mathbf{z}_t|\mathbf{y})} \left[D_{\mathrm{KL}}\left(\tilde{p}_{\boldsymbol{\alpha}}(\mathbf{z}_0|\mathbf{z}_t,\mathbf{y}) \,\|\, q(\mathbf{z}_0|\mathbf{z}_t,\mathbf{y})\right)\right]. \quad (20)$$

This establishes the inequality stated in the theorem. $\qquad \square$

**Full Statement of Lemma 3.2**

**Lemma B.2** (Full version of Lemma 3.2). *The KL divergence between the variational distribution* $\tilde{p}_{\boldsymbol{\alpha}}(\mathbf{z}_0|\mathbf{z}_t, \mathbf{y})$ *and the true posterior* $q(\mathbf{z}_0|\mathbf{z}_t, \mathbf{y})$ *can be decomposed into two terms:*

$$D_{\mathrm{KL}}\left(\tilde{p}_{\boldsymbol{\alpha}}(\mathbf{z}_0|\mathbf{z}_t, \mathbf{y})\|q(\mathbf{z}_0|\mathbf{z}_t, \mathbf{y})\right) = D_{\mathrm{KL}}\left(\tilde{p}_{\boldsymbol{\alpha}}(\mathbf{z}_0|\mathbf{z}_t, \mathbf{y})\|q(\mathbf{z}_0|\mathbf{z}_t)\right) - \mathbb{E}_{\mathbf{z}_0 \sim \tilde{p}_{\boldsymbol{\alpha}}(\mathbf{z}_0|\mathbf{z}_t, \mathbf{y})}\left[\log q(\mathbf{y}|\mathbf{z}_0)\right],$$
(21)

*where the first term in the right-hand side represents the KL divergence between the variational distribution and the prior distribution without the measurement condition, and the second term is the expected value of the negative log-likelihood* $-\log q(\mathbf{y}|\mathbf{z}_0)$ *under the variational distribution.*

*Proof.* We begin by considering the KL divergence between the variational distribution $\tilde{p}_{\boldsymbol{\alpha}}(\mathbf{z}_0|\mathbf{z}_t, \mathbf{y})$ and the true posterior $q(\mathbf{z}_0|\mathbf{z}_t, \mathbf{y})$. Given that $\mathbf{z}_0$ is a discrete variable, the KL divergence can be expressed as a sum:

$$D_{\mathrm{KL}}\left(\tilde{p}_{\boldsymbol{\alpha}}(\mathbf{z}_0|\mathbf{z}_t, \mathbf{y})\|q(\mathbf{z}_0|\mathbf{z}_t, \mathbf{y})\right) = \sum_{\mathbf{z}_0} \tilde{p}_{\boldsymbol{\alpha}}(\mathbf{z}_0|\mathbf{z}_t, \mathbf{y}) \log \frac{\tilde{p}_{\boldsymbol{\alpha}}(\mathbf{z}_0|\mathbf{z}_t, \mathbf{y})}{q(\mathbf{z}_0|\mathbf{z}_t, \mathbf{y})}.$$
(22)

By applying Bayes' theorem to the true posterior $q(\mathbf{z}_0|\mathbf{z}_t, \mathbf{y})$, we have:

$$q(\mathbf{z}_0|\mathbf{z}_t, \mathbf{y}) = \frac{q(\mathbf{z}_0|\mathbf{z}_t)q(\mathbf{y}|\mathbf{z}_0)}{q(\mathbf{y}|\mathbf{z}_t)}.$$
(23)

Since $q(\mathbf{y}|\mathbf{z}_t)$ does not depend on $\mathbf{z}_0$, it can be treated as a constant and ignored in the KL divergence calculation. Substituting Eq. (23) into Eq. (22), we obtain:

$$D_{\mathrm{KL}}\left(\tilde{p}_{\boldsymbol{\alpha}}(\mathbf{z}_0|\mathbf{z}_t, \mathbf{y})\|q(\mathbf{z}_0|\mathbf{z}_t, \mathbf{y})\right) = \sum_{\mathbf{z}_0} \tilde{p}_{\boldsymbol{\alpha}}(\mathbf{z}_0|\mathbf{z}_t, \mathbf{y}) \log \frac{\tilde{p}_{\boldsymbol{\alpha}}(\mathbf{z}_0|\mathbf{z}_t, \mathbf{y})}{q(\mathbf{z}_0|\mathbf{z}_t)q(\mathbf{y}|\mathbf{z}_0)}.$$
(24)

Next, we split the logarithm in the numerator and denominator:

$$D_{\mathrm{KL}}\left(\tilde{p}_{\boldsymbol{\alpha}}(\mathbf{z}_0|\mathbf{z}_t, \mathbf{y})\|q(\mathbf{z}_0|\mathbf{z}_t, \mathbf{y})\right) = \sum_{\mathbf{z}_0} \tilde{p}_{\boldsymbol{\alpha}}(\mathbf{z}_0|\mathbf{z}_t, \mathbf{y}) \left[\log \frac{\tilde{p}_{\boldsymbol{\alpha}}(\mathbf{z}_0|\mathbf{z}_t, \mathbf{y})}{q(\mathbf{z}_0|\mathbf{z}_t)} - \log q(\mathbf{y}|\mathbf{z}_0)\right].$$
(25)

This expression can be decomposed into two terms:

1. The first term represents the KL divergence between the variational distribution $\tilde{p}_{\boldsymbol{\alpha}}(\mathbf{z}_0|\mathbf{z}_t, \mathbf{y})$ and the prior $q(\mathbf{z}_0|\mathbf{z}_t)$:

$$D_{\mathrm{KL}}\left(\tilde{p}_{\boldsymbol{\alpha}}(\mathbf{z}_0|\mathbf{z}_t, \mathbf{y})\|q(\mathbf{z}_0|\mathbf{z}_t)\right) = \sum_{\mathbf{z}_0} \tilde{p}_{\boldsymbol{\alpha}}(\mathbf{z}_0|\mathbf{z}_t, \mathbf{y}) \log \frac{\tilde{p}_{\boldsymbol{\alpha}}(\mathbf{z}_0|\mathbf{z}_t, \mathbf{y})}{q(\mathbf{z}_0|\mathbf{z}_t)}.$$
(26)

2. The second term is the negative expected log-likelihood under the variational distribution:

$$\mathbb{E}_{\mathbf{z}_0 \sim \tilde{p}_{\boldsymbol{\alpha}}(\mathbf{z}_0|\mathbf{z}_t, \mathbf{y})}\left[-\log q(\mathbf{y}|\mathbf{z}_0)\right] = -\sum_{\mathbf{z}_0} \tilde{p}_{\boldsymbol{\alpha}}(\mathbf{z}_0|\mathbf{z}_t, \mathbf{y}) \log q(\mathbf{y}|\mathbf{z}_0).$$
(27)

Thus, the KL divergence between $\tilde{p}_{\boldsymbol{\alpha}}(\mathbf{z}_0|\mathbf{z}_t, \mathbf{y})$ and $q(\mathbf{z}_0|\mathbf{z}_t, \mathbf{y})$ can be decomposed as follows:

$$D_{\mathrm{KL}}\left(\tilde{p}_{\boldsymbol{\alpha}}(\mathbf{z}_0|\mathbf{z}_t, \mathbf{y})\|q(\mathbf{z}_0|\mathbf{z}_t, \mathbf{y})\right) = D_{\mathrm{KL}}\left(\tilde{p}_{\boldsymbol{\alpha}}(\mathbf{z}_0|\mathbf{z}_t, \mathbf{y})\|q(\mathbf{z}_0|\mathbf{z}_t)\right) - \mathbb{E}_{\mathbf{z}_0 \sim \tilde{p}_{\boldsymbol{\alpha}}(\mathbf{z}_0|\mathbf{z}_t, \mathbf{y})}\left[\log q(\mathbf{y}|\mathbf{z}_0)\right].$$
(28)

This concludes the proof. $\qquad\square$

**Lemma B.3.** *The marginal distribution $q_{sampling}(\mathbf{z}_0|\mathbf{y})$ is identical to the target distribution $q(\mathbf{z}_0|\mathbf{y})$.*

*Proof.* We aim to show that:

$$q_{\text{sampling}}(\mathbf{z}_0|\mathbf{y}) = q(\mathbf{z}_0|\mathbf{y}). \tag{29}$$

Starting from the definition of $q_{\text{sampling}}$:

$$q_{\text{sampling}}(\mathbf{z}_{0:T}|\mathbf{y}) = q(\mathbf{z}_T|\mathbf{y}) \prod_{t=1}^{T} q(\mathbf{z}_{t-1}|\mathbf{z}_t, \mathbf{y}). \tag{30}$$

Marginalizing over $\mathbf{z}_{1:T}$:

$$q_{\text{sampling}}(\mathbf{z}_0|\mathbf{y}) = \sum_{\mathbf{z}_{1:T}} q(\mathbf{z}_T|\mathbf{y}) \prod_{t=1}^{T} q(\mathbf{z}_{t-1}|\mathbf{z}_t, \mathbf{y}). \tag{31}$$

Since the forward process completely corrupts $\mathbf{z}_0$, we have $q(\mathbf{z}_T|\mathbf{z}_0) = q(\mathbf{z}_T)$, making $\mathbf{z}_T$ independent of $\mathbf{z}_0$. Consequently, $q(\mathbf{z}_T|\mathbf{y}) = q(\mathbf{z}_T)$ because $\mathbf{z}_T$ is independent of $\mathbf{y}$. Therefore:

$$q_{\text{sampling}}(\mathbf{z}_0|\mathbf{y}) = \sum_{\mathbf{z}_{1:T}} q(\mathbf{z}_T) \prod_{t=1}^{T} q(\mathbf{z}_{t-1}|\mathbf{z}_t, \mathbf{y}). \tag{32}$$

Since $q(\mathbf{z}_T)$ is constant with respect to $\mathbf{z}_0$ and $\mathbf{y}$, it can be factored out:

$$q_{\text{sampling}}(\mathbf{z}_0|\mathbf{y}) = q(\mathbf{z}_T) \sum_{\mathbf{z}_{1:T}} \prod_{t=1}^{T} q(\mathbf{z}_{t-1}|\mathbf{z}_t, \mathbf{y}). \tag{33}$$

The sum over $\mathbf{z}_{1:T}$ of the product of reverse transitions represents the total probability of generating $\mathbf{z}_0$ from any $\mathbf{z}_T$ using the reverse process conditioned on $\mathbf{y}$. Since $\mathbf{z}_T$ is sampled independently of $\mathbf{y}$ and $\mathbf{z}_0$, the reverse process effectively generates $\mathbf{z}_0$ solely based on $\mathbf{y}$. Therefore, the distribution over $\mathbf{z}_0$ is determined entirely by the reverse process:

$$q_{\text{sampling}}(\mathbf{z}_0|\mathbf{y}) = q(\mathbf{z}_T) \cdot (\text{probability of generating } \mathbf{z}_0 \text{ from reverse process}). \tag{34}$$

Since $q(\mathbf{z}_T)$ is a normalization constant and the reverse process is designed to sample $\mathbf{z}_0$ from $q(\mathbf{z}_0|\mathbf{y})$, we conclude:

$$q_{\text{sampling}}(\mathbf{z}_0|\mathbf{y}) = q(\mathbf{z}_0|\mathbf{y}). \tag{35}$$

$\square$

## C    DETAILS ON EXPERIMENTS

### C.1    IMAGE INVERSE PROBLEMS

**Implementation of Forward Operators and Dataset Selection**    In our image inverse problem experiments, the definition and implementation of the forward operator are based on the DPS implementation[2]. To ensure a diverse representation of ImageNet classes without genre bias, we select a subset consisting of 100 images from classes $0, 10, \ldots, 990$ using the `imagenet_val_1k.txt` provided by Pan et al. (2021)[3]. For our experiments with the FFHQ dataset, we use images $0, 1, \ldots, 99$ from the validation set.

---

[2] https://github.com/DPS2022/diffusion-posterior-sampling
[3] https://github.com/XingangPan/deep-generative-prior/

C.2 IMPLEMENTATION DETAILS OF G2D2 IN INVERSE PROBLEM SETTINGS

The implementation of G2D2 is based on the VQ-Diffusion model from the `diffusers` library [4]. For the prior model, we use the pre-trained model available at `https://huggingface.co/microsoft/vq-diffusion-ithq`. In our experiments, the number of time steps $T$ for sampling is set to 100.

**Parameterization of Star-Shaped Noise Process** In G2D2, the star-shaped noise process follows the same cumulative transition probability $q(\mathbf{z}_t|\mathbf{z}_0)$ as the original Markov noise process. For the Markov noise forward process where $q(\mathbf{z}_t|\mathbf{z}_{t-1})$ is defined using $Q_t$ as in Equation 2, the cumulative transition probability is computed as $q(z_{t,i}|\mathbf{z}_0) = \boldsymbol{v}^{\mathsf{T}}(z_{t,i})\overline{Q}_t\boldsymbol{v}(z_{0,i})$, where $\overline{Q}_t = Q_t \cdots Q_1$. Here, $\overline{Q}_t$ can be computed in closed form as:

$$\overline{Q}_t \boldsymbol{v}(z_{0,i}) = \overline{\alpha}_t \boldsymbol{v}(z_{0,i}) + (\overline{\gamma}_t - \overline{\beta}_t)\boldsymbol{v}(K+1) + \overline{\beta}_t, \tag{36}$$

where $\overline{\alpha}_t = \prod_{i=1}^{t-1}\alpha_i$, $\overline{\gamma}_t = 1 - \prod_{i=1}^{t-1}(1-\gamma_i)$, and $\overline{\beta}_t = (1 - \overline{\alpha}_t - \overline{\gamma}_t)/(K+1)$. These parameters can be calculated and stored in advance. The parameter settings follow those used during the training of the prior model. Specifically, $\overline{\alpha}_1$ is set to 0.99999, $\overline{\alpha}_T$ to 0.000009, $\overline{\gamma}_1$ to 0.000009, and $\overline{\gamma}_T$ to 0.99999. For both $\overline{\alpha}_t$ and $\overline{\gamma}_t$, values are linearly interpolated between steps 1 and $T$. This scheduling results in a linear increase in the number of [MASK] states as $t$ increases, ultimately leading to all variables transitioning to the [MASK] state. Additionally, the transition probability $\beta_t$ between unmasked tokens is set to be negligibly small, as $\overline{\alpha}_t$ and $\overline{\gamma}_t$ sum to nearly 1.

**Optimization in the Algorithm and Instantiation of the Objective Function** In the continuous optimization phase, we optimize the parameters $\boldsymbol{\alpha}$ of the categorical distribution using the Adam optimizer. The optimization objective is a weighted sum of the KL divergence term and the likelihood term, defined as:

$$\boldsymbol{\alpha}_t = \arg\min_{\boldsymbol{\alpha}_t} \left\{ \eta_{\mathrm{KL}} D_{\mathrm{KL}}\left(\tilde{p}_{\boldsymbol{\alpha}}(\mathbf{z}_0|\mathbf{z}_t,\mathbf{y})\|\tilde{p}_\theta(\mathbf{z}_0|\mathbf{z}_t)\right) + \|\mathbf{y} - \mathbf{A}\mathbf{x}_0(\boldsymbol{\alpha}_t)\|_2 \right\}, \tag{37}$$

where $\eta_{\mathrm{KL}}$ controls the trade-off between the KL term and the likelihood term.

**Marginalization over $\mathbf{z}_0$ in Algorithm 1** The marginalization over $\mathbf{z}_0$ in line 10 of Algorithm 1, specifically the term $\sum_{\mathbf{z}_0} q(\mathbf{z}_{t-1}|\mathbf{z}_0)\tilde{p}_{\boldsymbol{\alpha}}(\mathbf{z}_0|\mathbf{z}_t,\mathbf{y})$, can be computed in closed form. This computation is feasible because both distributions involved in the marginalization are dimensionally independent categorical distributions, as discussed by Austin et al. (2021) and Gu et al. (2022).

**Dynamic Learning Rate and KL Coefficient Scheduling** Some parameters are dynamically adjusted during inference. Both the learning rate for Adam ($l_{\mathrm{Adam}}$) and the KL divergence coefficient ($\eta_{\mathrm{KL}}$) are scheduled using weight vectors that decay logarithmically over the inference steps. These weights are computed based on initial scaling factors.

The learning rate weight vector $w_{\mathrm{lr}}$ and the KL coefficient weight vector $w_{\mathrm{KL}}$ are defined as follows:

$$w_{\mathrm{lr}}(t) = 10^{\left(\frac{\lambda_{\mathrm{lr,\,schedule}}}{2} \cdot \left(\frac{2t}{T}-1\right)\right)},$$

$$w_{\mathrm{KL}}(t) = 10^{\left(\frac{\lambda_{\mathrm{KL,\,schedule}}}{2} \cdot \left(\frac{2t}{T}-1\right)\right)}.$$

Here, $\lambda_{\mathrm{lr,\,schedule}}$ and $\lambda_{\mathrm{KL,\,schedule}}$ represent the initial scaling factors for the learning rate and KL coefficient, respectively, and $T$ is the total number of inference steps. When $\lambda_{\mathrm{lr,\,schedule}} > 0$, the learning rate weight vector $w_{\mathrm{lr}}(t)$ starts with relatively large values when $t$ is large and decays exponentially as $t$ decreases. Specifically, $w_{\mathrm{lr}}(t)$ reaches its minimum near $t = 1$ and its maximum near $t = T$. This scheduling enables stronger optimization during the initial inference steps, with the learning rate gradually decreasing in the later steps.

At each step $t$, the parameters are set as follows:

$$l_{\mathrm{Adam}}(t) = l_{\mathrm{Adam,\,base}} \cdot w_{\mathrm{lr}}(t), \quad \eta_{\mathrm{KL}}(t) = \eta_{\mathrm{KL,\,base}} \cdot w_{\mathrm{KL}}(t).$$

---

[4] `https://huggingface.co/docs/diffusers/main/en/api/pipelines/vq_diffusion`

**Task-Specific and Common Hyperparameters**  The hyperparameters for Gaussian deblurring and super-resolution tasks used in the experiments are shown in Table 3.

| Dataset | Task | Hyperparameter | Value |
|---|---|---|---|
| ImageNet | Gaussian Deblurring | $\eta_{\text{KL, base}}$ | 0.0003 |
| | | $\lambda_{\text{KL, schedule}}$ | 2.0 |
| | | $l_{\text{Adam, base}}$ | 1.0 |
| | | $\lambda_{\text{lr, schedule}}$ | 0.5 |
| ImageNet | Super-resolution | $\eta_{\text{KL, base}}$ | 0.0001 |
| | | $\lambda_{\text{KL, schedule}}$ | 2.0 |
| | | $l_{\text{Adam, base}}$ | 0.5 |
| | | $\lambda_{\text{lr, schedule}}$ | 2.0 |
| FFHQ | Gaussian Deblurring | $\eta_{\text{KL, base}}$ | 0.0003 |
| | | $\lambda_{\text{KL, schedule}}$ | 2.0 |
| | | $l_{\text{Adam, base}}$ | 0.5 |
| | | $\lambda_{\text{lr, schedule}}$ | 0.5 |
| FFHQ | Super-resolution | $\eta_{\text{KL, base}}$ | 0.0003 |
| | | $\lambda_{\text{KL, schedule}}$ | 2.0 |
| | | $l_{\text{Adam, base}}$ | 0.3 |
| | | $\lambda_{\text{lr, schedule}}$ | 2.0 |

Table 3: Hyperparameters for Gaussian Deblurring and Super-resolution tasks on ImageNet and FFHQ datasets.

The following hyperparameters are shared across all experiments: The number of iterations for the optimization is set to 30, the temperature for Gumbel-Softmax relaxation is 1.0, and the forget coefficient is 0.3. For the classifier-free guidance scale, we use 5.0 in ImageNet experiments and 3.0 in FFHQ experiments.

### C.3  G2D2 WITH MARKOV NOISE PROCESS

As discussed in Section 4.3, a variant of G2D2 can be derived by introducing the original Markov noise process in the graphical model. In that case, the algorithm is shown in Algorithm 2. The key point here is that the $q_{\text{Markov}}(\mathbf{z}_{t-1}|\mathbf{z}_0, \mathbf{z}_t)$ part is identical to that of the original Markov noise process, which is expressed as

$$q_{\text{Markov}}(z_{t-1,i}|\mathbf{z}_0, \mathbf{z}_t) = \frac{(\boldsymbol{v}^{\mathsf{T}}(z_{t,i})Q_t\boldsymbol{v}(z_{t-1,i}))(\boldsymbol{v}^{\mathsf{T}}(z_{t-1,i})\overline{Q}_{t-1}\boldsymbol{v}(z_{0,i}))}{\boldsymbol{v}^{\mathsf{T}}(z_{t,i})\overline{Q}_t\boldsymbol{v}(z_{0,i})}. \tag{38}$$

In the mask-absorbing type of Markov noise process, this posterior distribution does not revert tokens that have once become unmasked states back to masked tokens. As a result, it becomes difficult to correct errors that occur in the early stages of sampling in subsequent steps.

---

**Algorithm 2 G2D2 with Markov Noise Process**

---

**Require:** Input condition $\mathbf{y}$, pre-trained discrete diffusion model $p_\theta$, forget coefficient $\gamma$
1:  $\mathbf{z}_T \sim q(\mathbf{z}_T)$
2:  **for** $t = T, \ldots, 1$ **do**
3:      **if** $t = T$ **then**
4:          Initialize: $\boldsymbol{\alpha}_t = \log \tilde{p}_\theta(\mathbf{z}_0|\mathbf{z}_t)$
5:      **else**
6:          Initialize: $\boldsymbol{\alpha}_t = \exp(\gamma \log \boldsymbol{\alpha}_{t+1} + (1 - \gamma) \log \tilde{p}_\theta(\mathbf{z}_0|\mathbf{z}_t))$
7:      **end if**
8:      // continuous optimization
9:      $\boldsymbol{\alpha}_t = \arg\min_{\boldsymbol{\alpha}_t} D_{\text{KL}}\left(\tilde{p}_{\boldsymbol{\alpha}}(\mathbf{z}_0|\mathbf{z}_t, \mathbf{y})\|\tilde{p}_\theta(\mathbf{z}_0|\mathbf{z}_t)\right) - \mathbb{E}_{\mathbf{z}_0 \sim \tilde{p}_{\boldsymbol{\alpha}}(\mathbf{z}_0|\mathbf{z}_t, \mathbf{y})}\left[\log q(\mathbf{y}|\mathbf{z}_0)\right]$
10:     Sample $\mathbf{z}_{t-1} \sim p_{\boldsymbol{\alpha}}(\mathbf{z}_{t-1}|\mathbf{z}_t, \mathbf{y}) = \sum_{\mathbf{z}_0} q_{\text{Markov}}(\mathbf{z}_{t-1}|\mathbf{z}_0, \mathbf{z}_t)\tilde{p}_{\boldsymbol{\alpha}}(\mathbf{z}_0|\mathbf{z}_t, \mathbf{y})$
11:     // Note: The term $q_{\text{Markov}}(\mathbf{z}_{t-1}|\mathbf{z}_0, \mathbf{z}_t)$ uses the posterior distribution of the original Markov noise process.
12: **end for**
13: **return** $\mathbf{x}_0$ by decoding $\mathbf{z}_0$

---

## C.4 Settings for comparison methods

In this subsection, we detail the experimental settings for the comparison method.

**DPS**  (Chung et al., 2023b) We use the same parameter settings as described in the original paper. The guidance scale is set to 1.0 for FFHQ & Super resolution, 1.0 for FFHQ & Gaussian deblurring, 1.0 for ImageNet & Super resolution, and 0.4 for ImageNet & Gaussian deblurring. The number of time steps is set to 1000. For pre-trained models, we use the unconditional model provided by Dhariwal & Nichol (2021) [5] for ImageNet. For FFHQ, we use the model provided by Choi et al. (2021) [6].

**DDRM**  (Kawar et al., 2022) We use the official implementation [7]. The time steps are set to $T = 20$, with $\eta = 0.85$ and $\eta_b = 1.0$ as the hyperparameters. For ImageNet, we use the same pre-trained model as DPS. Although there is no official implementation using a pre-trained model trained on FFHQ, both DDRM and Choi et al. (2021) are based on the implementation of Dhariwal & Nichol (2021). Therefore, in our experiments, DDRM uses the same pre-trained model as DPS.

**PSLD**  (Rout et al., 2023) We use the official implementation [8]. For the pre-trained model, we employ stable-diffusion v-1.5 (Rombach et al., 2022) [9]. As this model handles 512×512 pixel images, we first upscale the ground truth image to 512×512. We then apply the forward operator to the upscaled image and use the result as observed data for our method. Finally, we downsample the output to 256×256. For hyperparameters, we use $\eta = 1.0$ and $\gamma = 0.1$.

**ReSample**  (Song et al., 2024) We use the official implementation [10]. For pre-trained models, we employ two models from the latent diffusion models repository [11]: LDM-VQ-4 trained on FFHQ, and LDM-VQ-8 trained on ImageNet with class conditioning. We use $T = 500$ DDIM steps with $\tau$ set to $10^{-4}$. The maximum number of optimization steps is set to 500. The variance hyperparameter $\gamma$ is set to 40. For the ImageNet experiments, we input the class labels of the ground truth data to the model.

## C.5 GPU memory usage and computational speed

We analyze the GPU memory consumption and computational speed of our proposed method, G2D2, in comparison with other methods. Table 4 presents a overview of these metrics for various methods. The measurement are conducted using a single NVIDIA A6000 GPU for the Gaussian deblurring task on ImageNet. G2D2 has the lowest memory usage among all methods and the fastest computational speed among gradient-based methods.

Table 4: Comparison of GPU Memory Usage and Computational Speed

| Method | GPU Memory Usage (GiB) | Computational Time (s) |
|---|---|---|
| G2D2 (Proposed) | 4.7 | 194 |
| DPS | 10.7 | 277 |
| DDRM | 5.8 | 4 |
| PSLD | 20.9 | 738 |
| ReSample | 7.1 | 555 |

## C.6 Impact of the Forget Coefficient

Figure 7 shows the reduction in the loss function and the final results for the Gaussian deblurring task on ImageNet when the forget coefficient is set to 0.3 and 1.0. The case with a forget coefficient

---

[5] https://github.com/openai/guided-diffusion
[6] https://github.com/jychoi118/ilvr_adm
[7] https://github.com/bahjat-kawar/ddrm
[8] https://github.com/LituRout/PSLD
[9] https://github.com/CompVis/stable-diffusion
[10] https://github.com/soominkwon/resample
[11] https://github.com/CompVis/latent-diffusion

of 1.0 corresponds to not using the optimization results from the previous step at all. Introducing the forget coefficient allows for a faster reduction in the loss function and achieves higher performance with the same computational resources.

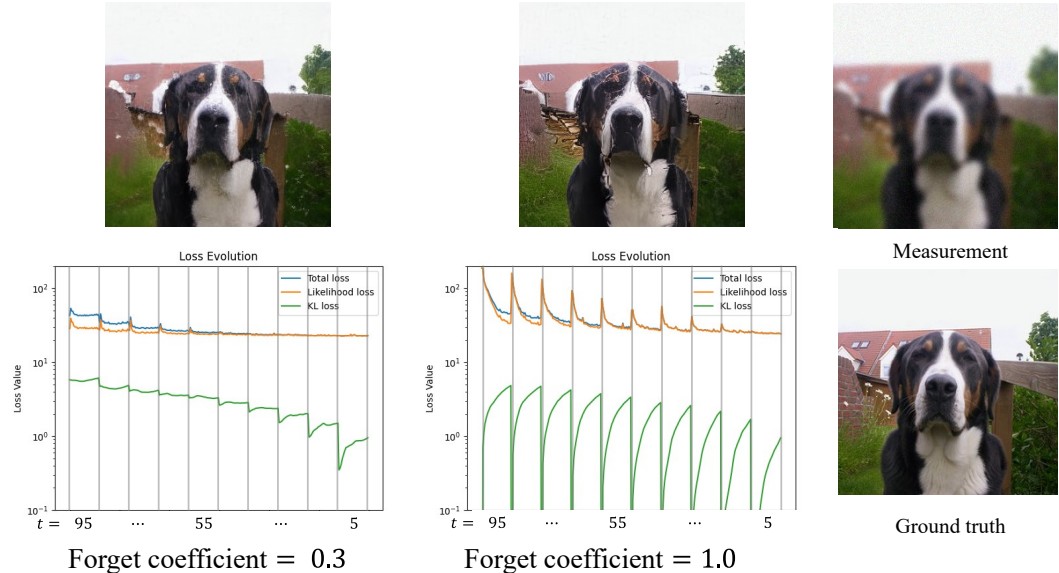

Forget coefficient = 0.3          Forget coefficient = 1.0

Figure 7: Reduction in the loss function and final results for the Gaussian deblurring task on ImageNet with forget coefficients of 0.3 and 1.0. The forget coefficient of 1.0 corresponds to not using the optimization results from the previous step.

## C.7 IMPACT OF TEXT CONDITIONING ON THE PRIOR MODEL

To examine the necessity of text conditioning, we investigate the effect of the presence or absence of prompts given to VQ-Diffusion on performance. Table 5 shows the performance for each setting. "Not Used" for text conditioning indicates that classifier-free guidance in the prior model is set to 1.0 (equivalent to unconditional sampling). The prompts we provide to VQ-Diffusion in our method are "a photo of [Class Name]" for ImageNet experiments and "a high-quality headshot of a person" for FFHQ experiments. It should be noted that these prompts are extremely general and do not describe specific details of the images. From these results, we can confirm that prompt conditioning contributes to a certain level of performance improvement.

| Dataset | Text conditioning | SR (×4) | | Gaussian Deblurring | |
|---------|-------------------|---------|---------|---------|---------|
| | | LPIPS↓ | PSNR↑ | LPIPS↓ | PSNR↑ |
| ImageNet | Not Used | 0.355 | **23.32** | 0.410 | 22.21 |
| | Used | **0.349** | 23.20 | **0.375** | **22.71** |
| FFHQ | Not Used | 0.300 | 26.60 | 0.328 | **25.43** |
| | Used | **0.271** | **26.93** | **0.288** | 24.42 |

Table 5: Performance comparison with and without text conditioning

## C.8 FAILURE MODES OF G2D2

We conduct an analysis of failure modes. Figure 8 shows the results of G2D2 and the images during inference for the Gaussian deblurring task on FFHQ. When the ground truth image is a relatively young (child's) face, the generated face images appear to be drawn towards a distribution of more adult faces. This is likely due to the use of the prompt "a high-quality headshot of a person". As a result, there is a consistent bias towards adult face images throughout the generation process, leading to artifacts in the final image. In the absence of a prompt, the intermediate generated images are not influenced by any specific textual guidance. As a result, the final image tends to have fewer artifacts.

While the star-shaped noise process can correct early errors, if errors persist until the later stages, it becomes more difficult to correct them from that point onwards. In other words, when there is a mismatch between the distribution conditioned by the prompt and the target image, it becomes challenging for G2D2 to handle it effectively.

To improve these issues, techniques such as simultaneous optimization of prompts may be necessary. Prompt-tuning techniques, as proposed in reference Chung et al. (2024), could be effective in addressing these challenges.

With prompt: "a high-quality headshot of a person"

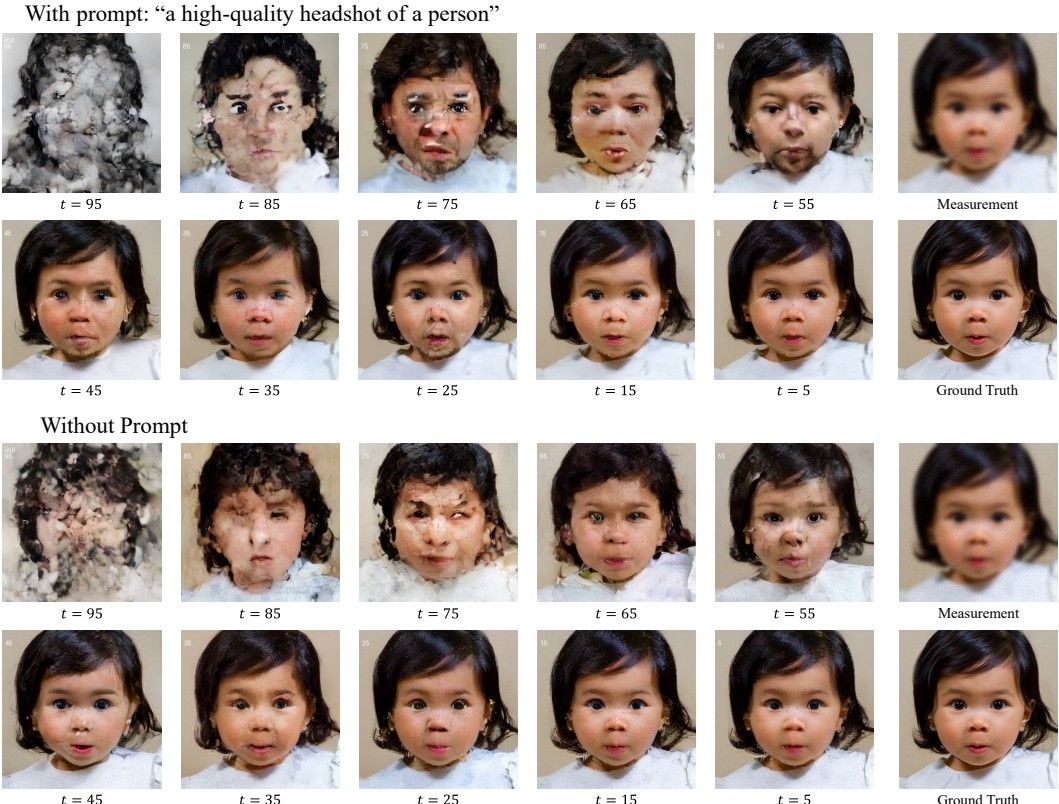

Figure 8: Failure modes of G2D2: Gaussian deblurring results on the FFHQ dataset. Due to the mismatch between the prompt and the target image, errors remain uncorrected throughout the process, resulting in artifacts in the estimated image.

## C.9    Additional qualitative results of G2D2 and comparison methods.

We present additional qualitative results of G2D2 and comparison methods. Figures 9 through 12 showcase the results for super-resolution (SR) and Gaussian blur (GB) tasks on ImageNet and FFHQ datasets.

## C.10    Additional qualitative results of G2D2 with Markov noise process

To compare G2D2 and G2D2 with Markov noise process, we present their respective qualitative results in Figures 13 and 14. The latter approach does not include re-masking operations in its sampling process, which means that once a token becomes unmasked, it cannot be modified in subsequent iterations. The unnatural artifacts observed in the resulting images are likely attributable to this limitation. This observation underscores the validity of adopting the star-shaped noise process in our proposed method.

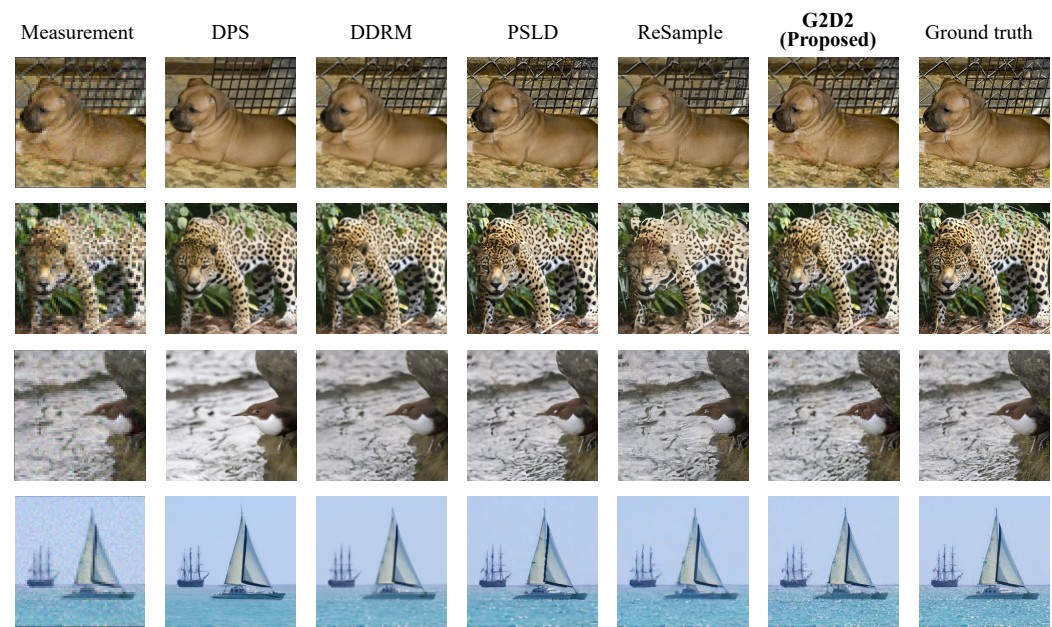

Figure 9: Qualitative results of G2D2 and comparison methods.

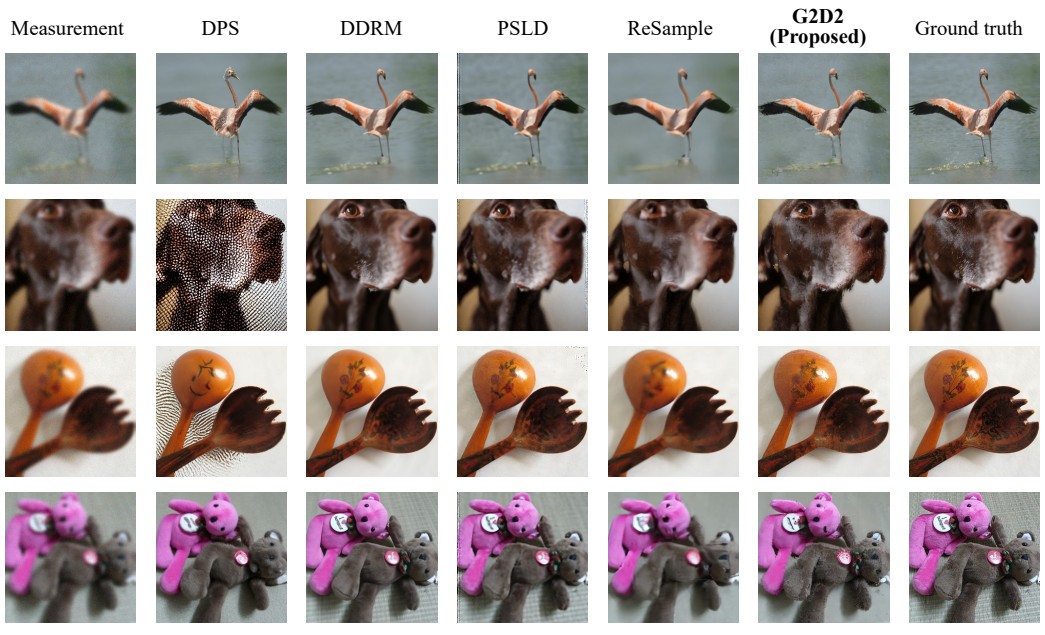

Figure 10: Qualitative results of G2D2 and comparison methods.

## C.11 INVERSE PROBLEMS ON MOTION DATA

We develop G2D2 based on the official implementation of MMM (Pinyoanuntapong et al., 2024) [12]. This method learns a masked generative model on the discrete latent space obtained by a motion

---

[12] https://github.com/exitudio/MMM

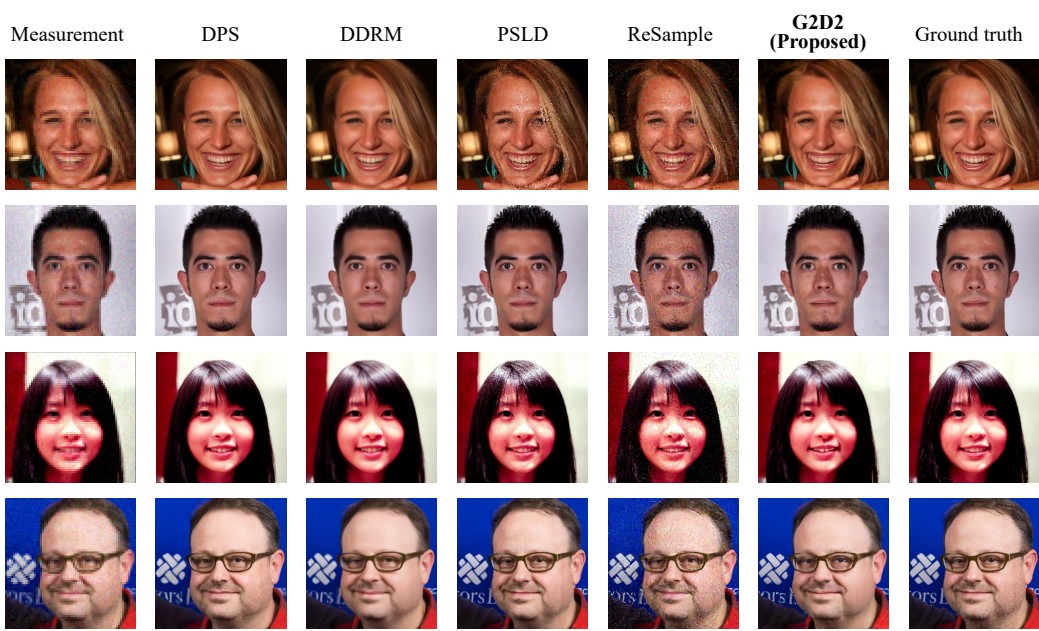

| Measurement | DPS | DDRM | PSLD | ReSample | G2D2 (Proposed) | Ground truth |

Super resolution (x4, $\sigma = 0.05$)

Figure 11: Qualitative results of G2D2 and comparison methods.

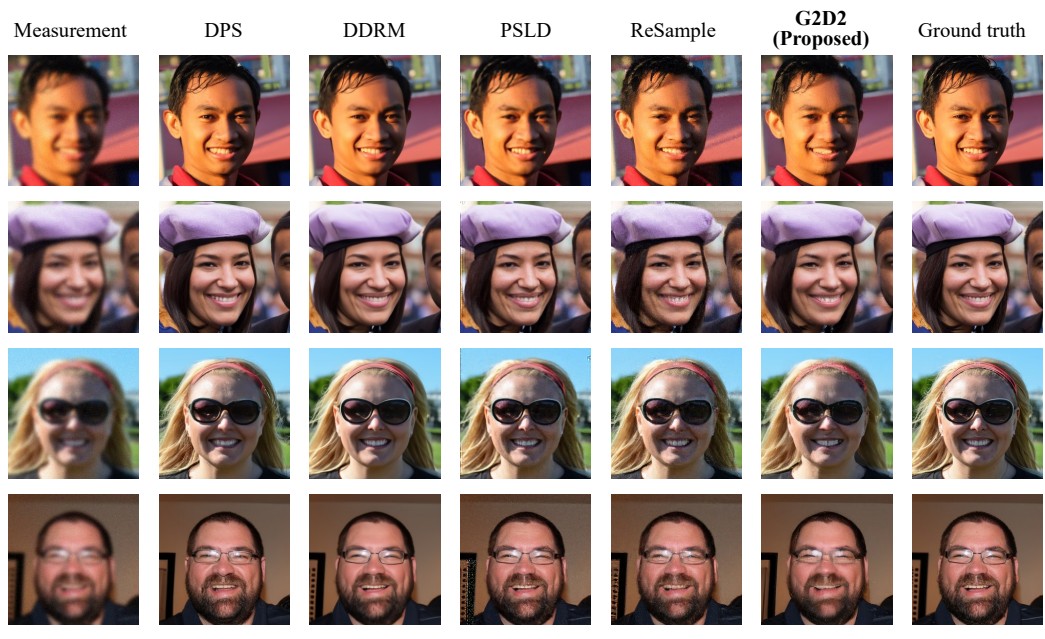

| Measurement | DPS | DDRM | PSLD | ReSample | G2D2 (Proposed) | Ground truth |

Gaussian Deblur ($\sigma = 0.05$)

Figure 12: Qualitative results of G2D2 and comparison methods.

tokenizer trained on the VQVAE framework (Van Den Oord et al., 2017). G2D2 uses the provided pre-trained model as a prior distribution.

We conduct experiments on the path following task (Song et al., 2023b; Uchida et al., 2024). The objective is to generate motion data $\mathbf{m}_0 \in \mathbb{R}^{d_{\mathbf{m}} \times L}$ that follows a given path $\mathbf{y}_{\text{path}} \in \mathbb{R}^{3 \times L}$. Here,

| Measurement | **G2D2 w/ Markov noise process** | **G2D2 (Proposed)** | Ground truth |
|---|---|---|---|

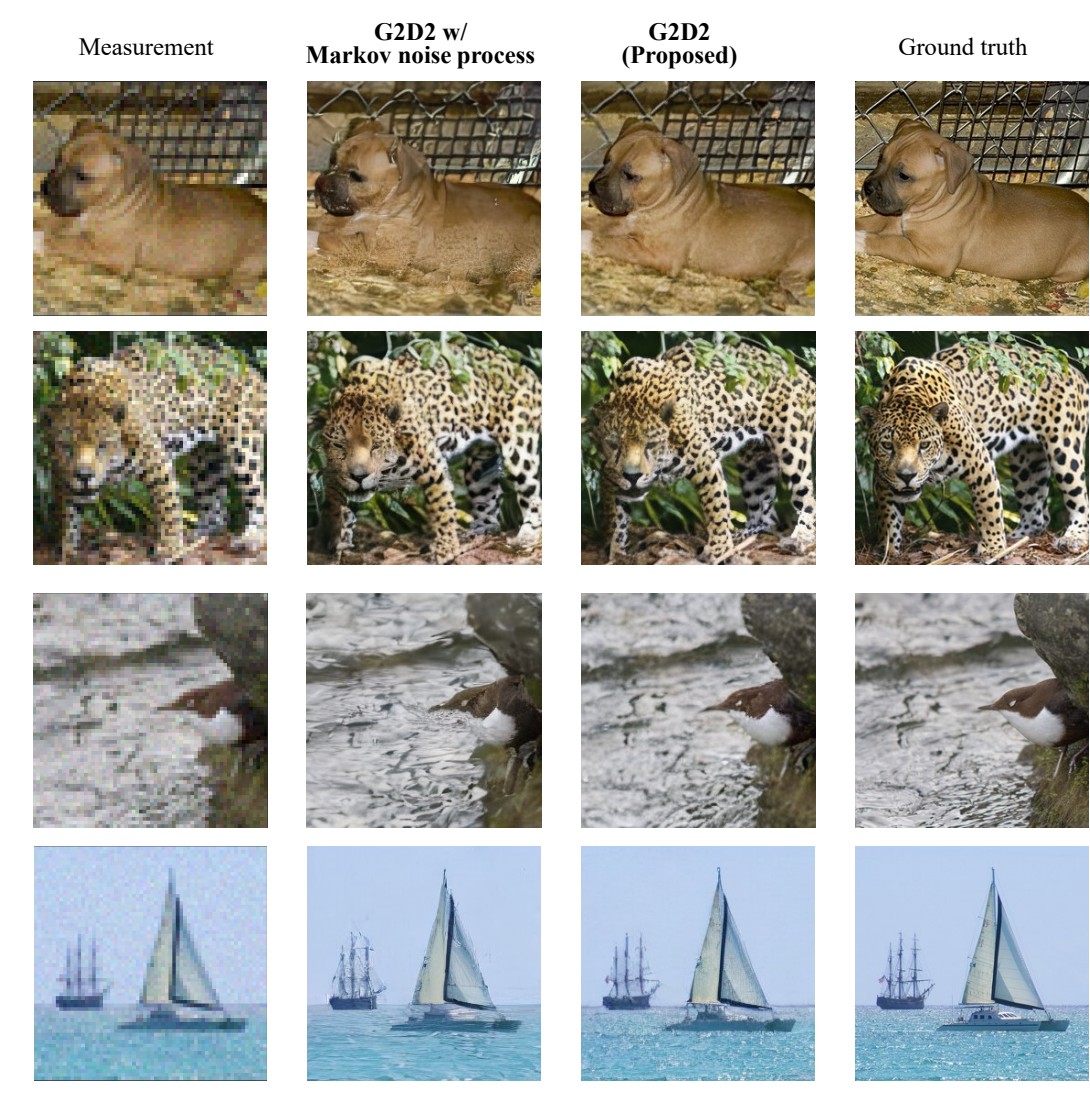

Super resolution (x4, $\sigma = 0.05$)

Figure 13: Qualitative results comparing G2D2 and G2D2 with Markov noise process (Super-resolution task).

$\mathbf{y}_{\text{path}}$ represents the coordinates of the hip joint at each time frame, $L$ denotes the number of frames in the motion data, and $d_{\mathbf{m}}$ is the dimensionality of each motion data point.

The likelihood loss used in the optimization process of G2D2 measures how closely the generated motion follows the target path. It is defined as

$$\log q(\mathbf{y}_{\text{path}}|\mathbf{m}_0) = \sum_{l=1}^{L} \|\mathbf{y}_{\text{path},l} - \mathbf{A}_{\text{path}}\mathbf{m}_{0,l}\|_2, \tag{39}$$

where $\mathbf{A}_{\text{path}}$ is a linear operator that extracts the path across the frames.

In our experiments, we use two types of paths (forward: a path moving forward at a constant speed, and zigzag: a path moving forward while zigzagging) and two prompts: "A person walks with his hands up" and "A person does a cart wheel." We also perform unconditional generation. The qualitative results are shown in Figure 15.

We conduct experiments with a total of $T = 25$ time steps. For hyperparameters, we set the number of iterations for optimization to 20 and the Gumbel-Softmax temperature to 1.0. The forget coeffi-

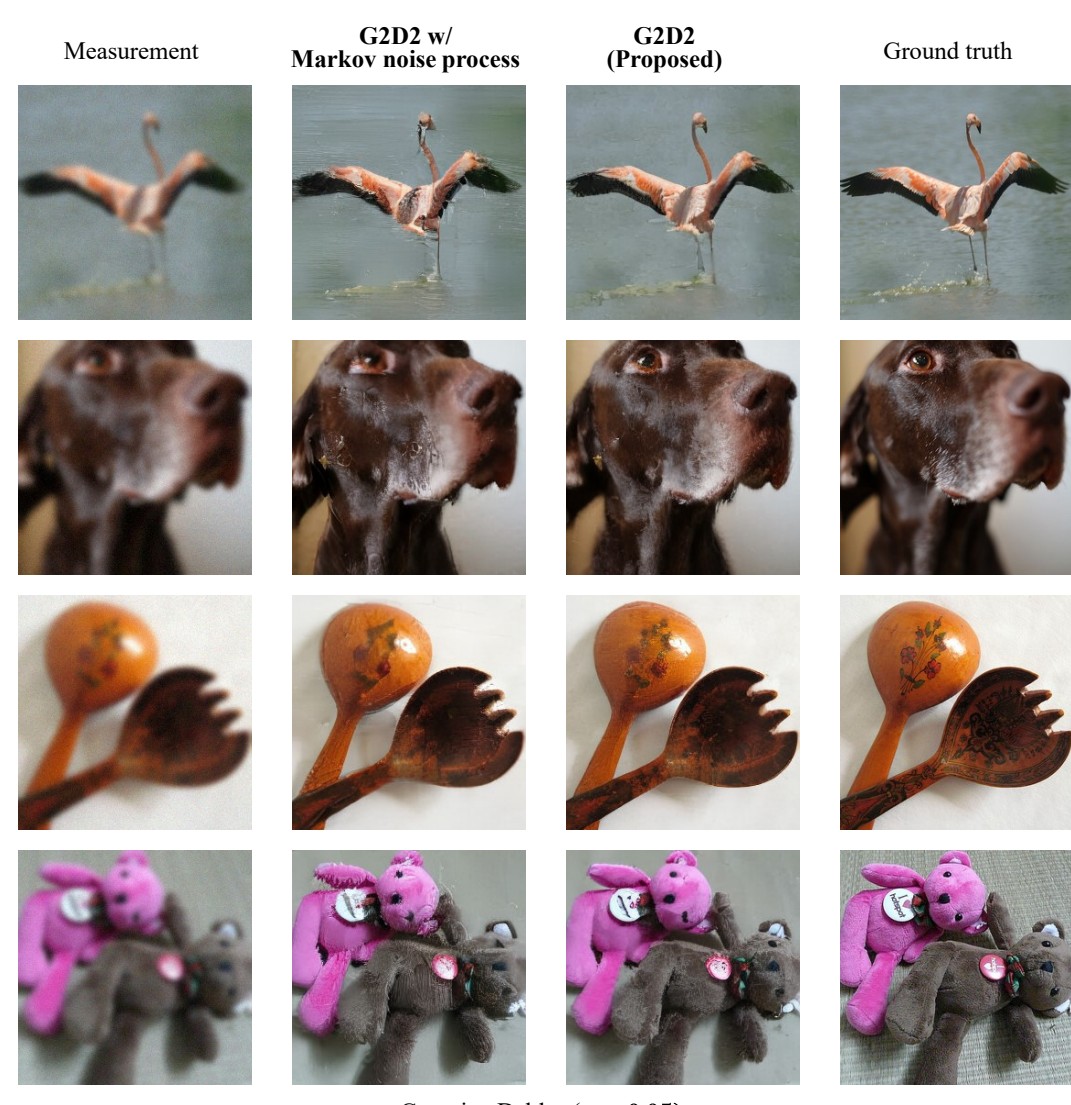

| Measurement | **G2D2 w/ Markov noise process** | **G2D2 (Proposed)** | Ground truth |
|---|---|---|---|

Gaussian Deblur ($\sigma = 0.05$)

Figure 14: Qualitative results comparing G2D2 and G2D2 with Markov noise process (Gaussian deblurring task).

cient is set to 0.7. We adopt the dynamic learning rate scheduling described in the Appendix C.2. The base Adam learning rate $l_{\text{Adam, base}}$ is set to 0.3, and the KL divergence weight $\eta_{\text{KL}}$ is set to 0.05. Additionally, we set $\lambda_{\text{KL, schedule}}$ and $\lambda_{\text{lr, schedule}}$ to 0.0 and 1.0, respectively.

## C.12    QUANTITATIVE COMPARISON WITH OTHER METHODS FOR CONTROLLABLE MOTION GENERATION

We conduct a comparison between G2D2 and existing methods in the controllable motion generation task. For comparison, we select OmniControl (Xie et al., 2024) and Guided Motion Diffusion (GMD) (Karunratanakul et al., 2023). The quantitative results are cited from their respective papers.

The comparison is performed on a path following task, specifically generating motion based on a prescribed trajectory for the pelvis. Following OmniControl's setup, we compare the following metrics under the sparse condition of 5 frames out of 196 frames in the HumanML3D test set: FID, R-Precision, Diversity, Foot Skating ratio, Trajectory error (50cm), Location error (50cm), and Average error.

The results are presented in the Table 6.

It's important to note that while OmniControl and GMD are fine-tuned for this specific task, G2D2 is a method that can be used without additional training. However, OmniControl and GMD show better performance in trajectory and location errors, suggesting that there is still room for improvement in the current version of G2D2.

We propose that methods like G2D2 can be used in combination with fine-tuned approaches. Additionally, there's potential to further enhance G2D2's performance by incorporating techniques such as the time-traveling method used in FreeDoM (Yu et al., 2023).

| Method | FID ($\downarrow$) | R-prec. ($\uparrow$) | Diversity (9.503$\rightarrow$) | Foot skating ($\downarrow$) | Traj. Err (50cm, $\downarrow$) | Loc. err. (50cm, $\downarrow$) | Avg. err. ($\downarrow$) |
|---|---|---|---|---|---|---|---|
| G2D2 | 0.248 | 0.770 | 9.381 | 0.048 | 0.272 | 0.116 | 0.230 |
| OmniControl (Xie et al., 2024) | 0.278 | 0.705 | 9.582 | 0.058 | 0.053 | 0.015 | 0.043 |
| GMD Karunratanakul et al. (2023) | 0.523 | 0.599 | N/A | 0.086 | 0.176 | 0.049 | 0.139 |

Table 6: Comparison of methods for controllable motion generation

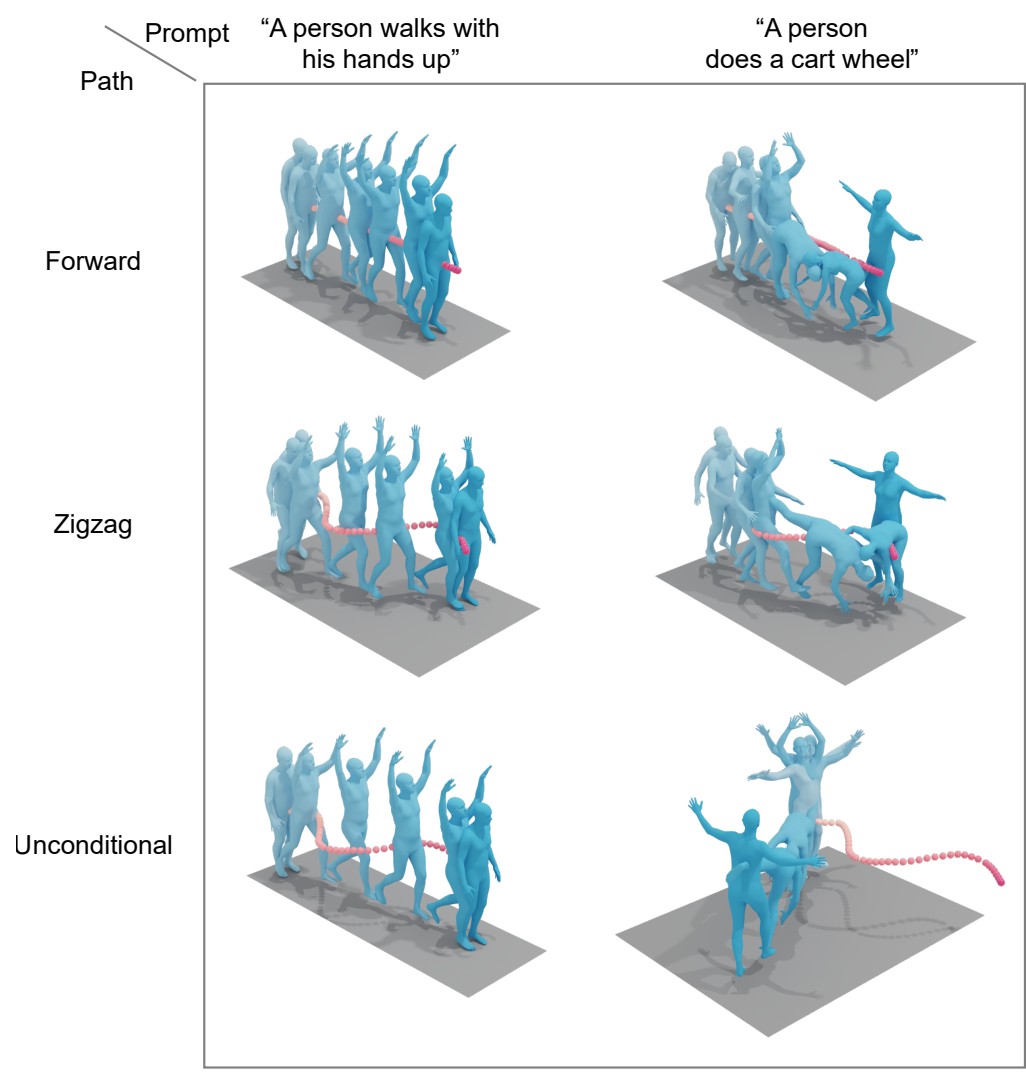

Figure 15: Qualitative results of motion inverse problem solving.

