# OpenReview forum: "G2D2: Gradient-guided Discrete Diffusion for image inverse problem solving"
_ICLR.cc/2025/Conference — Submitted to ICLR 2025_

### Official Review · Reviewer_GC4Q · 2024-10-15

**Soundness:** 3
**Presentation:** 2
**Contribution:** 2
**Rating:** 5
**Confidence:** 3

**Summary:**

This paper solves inverse problems in image processing by leveraging discrete diffusion models. Unlike prior approaches using continuous diffusion models, G2D2 introduces a star-shaped noise process and variational relaxation to enable the use of discrete diffusion models in continuous image reconstruction tasks, such as super-resolution and Gaussian deblurring.

**Strengths:**

1. G2D2 is original in its use of discrete diffusion models for solving inverse problems, which were previously limited to continuous domains. The introduction of the star-shaped noise process and continuous relaxation techniques shows a creative solution to a challenging problem of applying discrete models to continuous tasks.

**Weaknesses:**

1. While G2D2 introduces innovative techniques, the computational complexity of the variational optimization and continuous relaxation might be higher than other methods. The paper could benefit from more detailed discussions on the computational trade-offs and potential optimizations to make the approach more efficient.


2. The paper focuses primarily on image processing tasks such as super-resolution and deblurring, with limited exploration of other possible applications. Extending the method to more diverse and complex inverse problems, such as in other imaging modalities or higher-dimensional data (e.g., 3D medical imaging), would improve the robustness and generalizability of G2D2.

3. Although G2D2 is compared with several continuous diffusion models, a more thorough comparison with the latest emerging techniques, such as latent diffusion models or score-based generative models, would strengthen the evaluation.

4. The paper emphasizes that the star-shaped noise process can correct early-stage errors, but it does not explore the limitations of this mechanism in depth. There could be scenarios where certain types of errors are less tractable, and a discussion on the failure modes of G2D2


5. More related work should be cited such as "Diffusion modeling with domain-conditioned prior guidance for accelerated mri and qmri reconstruction".

**Questions:**

1. How does the star-shaped noise process enable the "re-masking" operation, and why is this beneficial for solving inverse problems?

2. What are the advantages of using a star-shaped noise process over a standard Markov process for sampling in discrete diffusion models?

3. What does the term "conditionally independent" imply about the relationship between the noisy discrete latents z1, z2,...,zT given z0 in the star-shaped noise process?

4. What does Property 2 suggest about the relationship between the marginal distributions of the joint distribution q_sampling and the star-shaped noise process graphical model?

5. How does the mean-field structure of both q(zt - 1 | z0) and p_\alpha (z0|zt,y) impact the marginalization process in the G2D2 method?

6. In the algorithm, What challenges might arise if the optimization process does not incorporate the values from the previous time step during initialization?

---

> ### Author Response · Authors · 2024-11-23
> **Rebuttal response to Reviewer GC4Q**
>
> Thank you for your valuable feedback. We will respond to your questions and concerns in the following sections.
>
> ### Computational Complexity and Optimization of G2D2 (Weakness 1)
>
> We agree that an analysis of computational complexity is important. In G2D2, the primary computational overhead is the optimization of the variational distribution. However, this overhead is common to gradient-based methods (such as DPS, PSLD, and ReSample), and in our experiments (aided by the introduction of a forgetting factor), it has not resulted in significant overhead. Rather, since we can skip gradient calculations within pre-trained prior models, unlike DPS and PSLD, we have advantages in terms of GPU memory usage and inference speed. On the other hand, as pointed out, there is still room for optimization in the optimization methods. For example, we currently use the same number of optimization steps for all steps (e.g., 30 steps for each time step), but according to Figure 7, some time steps converge faster. Properly scheduling these steps could lead to further acceleration of inference speed.
>
> ### Expanding Applications Beyond Image Processing (Weakness 2)
>
> Due to time constraints, our focus was on image tasks, but we recognize the importance of demonstrating versatility. In particular, applying our method to other image domains such as 3D medical imaging is of interest. While it could be possible to achieve similar results as in literature [7] using our method, it should be noted that there are currently no discrete diffusion models trained on available medical imaging data. Future developments in prior models may enable applications in these domains.
>
> [7] Yang, et al., "Solving inverse problems in medical imaging with score-based generative models", ICLR2022
>
> ### Comparison with Additional Baselines (Weakness 3)
>
> We agree with the suggestion that comparing with additional baselines would strengthen the claims in our paper. We have already conducted comparisons with methods using continuous diffusion models in the pixel domain (DPS, DDRM) and methods using latent diffusion models trained in the latent space (PSLD, ReSample). Additionally, we recognize that the mentioned score-based generative models belong to these groups. We believe these comparisons provide the necessary minimum validation.
>
> ### Failure modes of G2D2 (Weakness 4)
>
> We believe that analyzing failure modes can lead to constructive improvements in this method. Upon observing the G2D2's results in linear inverse problems, we discovered some interesting phenomena in the FFHQ results.. Please refer to Figure 8 in Appendix C.8. in the revised manuscript for the images. When the ground truth image is a relatively young (child's) face, the generated face images seem to be pulled towards a distribution of more adult faces, possibly due to the use of the prompt "a high-quality headshot of a person". This results in a consistent bias towards adult face images throughout the generation process, leading to artifacts in the final image. While the star-shaped noise process can correct early errors, if errors persist until the later stages, it becomes more difficult to correct them from that point onwards. In other words, when there is a mismatch between the distribution conditioned by the prompt and the target image, it becomes challenging for G2D2 to handle it effectively. To improve these issues, techniques such as simultaneous optimization of prompts may be necessary. Prompt-tuning techniques, as proposed in reference [8], could be effective in addressing these challenges.
>
> We have added this discussion to the revised manuscript in Appendix C.8..
>
> [8] Chung, et al., "Prompt-tuning Latent Diffusion Models for Inverse Problems", ICML2024.
>
> ### Related Works (Weakness 4)
>
> Thank you for pointing out the missing citation. We have added a section in the Appendix's Related Works that cites "Diffusion Modeling with Domain-Conditioned Prior Guidance for Accelerated MRI and qMRI Reconstruction" and other relevant studies.

---

> > ### Author Response · Authors · 2024-11-23
> > **(Cont.) Rebuttal response to Reviewer GC4Q**
> >
> > ### Responses to Reviewer GC4Q's Questions
> >
> > We would like to address the questions raised by Reviewer GC4Q as follows.
> >
> > ### How does the star-shaped noise process enable the "re-masking" operation, and why is this beneficial for solving inverse problems? (Question 1)
> >
> > ### What are the advantages of using a star-shaped noise process over a standard Markov process for sampling in discrete diffusion models? (Question 2)
> >
> > As mentioned in the Introduction and Section 3.1, the adoption of the star-shaped noise process eliminates the correlation of mask positions between steps, implicitly realizing the remasking operation. This allows tokens that have been unmasked once to transition to different tokens again, potentially correcting errors that occurred in the early stages of sampling. The Experiment section shows the quantitative effectiveness through comparative experiments between G2D2 and G2D2 w/ Markov noise process, and the qualitative comparison is presented in Figure 12. Please refer to this as well.
> >
> > ### What does the term "conditionally independent" imply about the relationship between the noisy discrete latents z1, z2,...,zT given z0 in the star-shaped noise process? (Question 3)
> >
> > The term "conditionally independent" means that given $z_0$, the $z_t$ corresponding to different $t$ are independent of each other. (Given $z_0$, $z_t$ provides no information about $z_{t'}$ corresponding to a different $t'$, and vice versa.) In the context of the mask-absorbing state type of discrete diffusion models we handle, this means that there is no correlation in the mask positions between $z_t$ and $z_{t'}$ at different time steps. On the other hand, in the Markov noise process, the positions of the [MASK] tokens increase as $t$ increases (the existing [MASK] tokens remain unchanged). Therefore, the mask positions are correlated between different time steps $t$ and $t'$.
> >
> > ### What does Property 2 suggest about the relationship between the marginal distributions of the joint distribution q_sampling and the star-shaped noise process graphical model? (Question 4)
> >
> > After defining the star-shaped noise process, the most straightforward way to achieve sampling from the pure $q(z_0|y)$ is to aim for sampling from $q(z_{0:T}|y)$. However, this requires considering the $z_t$ at multiple time steps simultaneously, which is not practical. By considering $q_{sampling}$, although its joint distribution differs from that of the star-shaped noise process (Property 3), we can see that focusing solely on the sampling of $z_0$ allows us to achieve sampling from the marginal distribution.
> >
> > ### How does the mean-field structure of both q(zt - 1 | z0) and p_α (z0|zt,y) impact the marginalization process in the G2D2 method? (Question 5)
> >
> > This indicates that $q(z_{t-1}|z_0)$ and $p_\alpha(z_0|z_t, y)$ both have the structure of dimensionally independent categorical distributions, making marginalization easy to compute. Specifically, generating a sequence of tokens from the dimensionally-wise independent categorical distribution of $p_\alpha(z_0|z_t, y)$ and then transitioning some tokens to [MASK] tokens according to $q(z_{t-1}|z_0)$ corresponds to sampling from the marginalized distribution.
> >
> > ### In the algorithm, What challenges might arise if the optimization process does not incorporate the values from the previous time step during initialization? (Question 6)
> >
> > Not implementing the optimization initialization strategy described in Section 3.3 leads to an inefficient optimization process. Specifically, as shown in Figure 7, insufficient optimization results in artifacts in the output images.

---

> > > ### Author Response · Authors · 2024-11-27
> > > **Looking forward to your response**
> > >
> > > Thank you for your thorough and insightful review. Although the rebuttal period has been extended, we are eager to address all of your concerns to the best of our ability. We would like to know if you have any remaining questions or issues that we haven't yet addressed. We look forward to your response. Thank you for your time and consideration.

---

### Official Review · Reviewer_ZmJs · 2024-10-18

**Soundness:** 3
**Presentation:** 2
**Contribution:** 3
**Rating:** 6
**Confidence:** 5

**Summary:**

The authors propose G2D2, a diffusion model-based inverse problem solver (DIS) that uses a discrete diffusion model. To the best of my knowledge, this is the first work to demonstrate that this is possible. Since using the usual mask-observing Markov process of discrete diffusion models makes it hard to correct for the errors arising in the earlier stages of the sampling, the authors propose to use a star-shaped diffusion, where similar to DDIM, $z_t$s are conditionally independent given $z_0$. During inference, the parameters $\alpha$ of the variational reverse categorical distribution are optimized by balancing the prior and the likelihood, which is grounded by sound theory. Experiments are conducted on a standard FFHQ/ImageNet settings.

**Strengths:**

1. To the best of my knowledge, this is the first work to target using discrete diffusion models for solving inverse problems. G2D2 will open up new opportunities for testing out different discrete diffusion priors.

2. The theory is sound, and the resulting algorithm is straightforward to understand and implement. This resembles how most of the current DIS is implemented in practice, where the predicted $x_0$ is used to compute the likelihood loss, and the sampling to $x_{t-1}$ is conducted with a DDIM sampling step.

**Weaknesses:**

1. The results are weak. This is somewhat understandable given that the pre-trained diffusion prior is suboptimal, and latent diffusion models (whether continuous or discrete) tend to be inferior to pixel-based methods due to the existence of decoders.

2. The presentation could be improved. Using a star-shaped noise process is one of the crucial contributions of the work, but this is first introduced in 3.1. A brief review of this before the main section would be beneficial for better understanding.

3. Many readers interested in the work will already be familiar with the family of continuous diffusion-based methods, and in many points, G2D2 resembles them. It would be beneficial for the authors to draw links to the continuous counterpart, especially in the construction of the sampling.

4. Motion inverse problem solving is demonstrated without any comparison.

5. It is discussed that G2D2 can be used with MaskGIT, which is one of the main strength of the work. However, MaskGIT is only used for motion inverse problem solving, which only consists of a small proportion of the experiments.

**Questions:**

Did the authors try using MaskGIT prior to image inverse problems? It is a better prior than VQ-diffusion, and it would be surprising if the results did not improve by simply switching the pre-trained model.

---

> ### Author Response · Authors · 2024-11-23
> **Rebuttal response to Reviewer ZmJs**
>
> We are grateful for your constructive feedback and recognition of our efforts. Below, we will address the questions and concerns you have raised.
>
> ### Presentation of the Star-shaped Noise Process (Weakness 1)
>
> Thank you for the suggestions for improvement. Indeed, the current manuscript lacked sufficient description of the star-shaped noise process in the Introduction. We have now enhanced the description of the Star-shaped noise process in lines 99-107.
>
> ### Connections to the Continuous Diffusion-based Methods (Weakness 2)
>
> We agree that emphasizing the connections to continuous diffusion methods is important. We will add a comparative analysis to the Appendix of the manuscript that outlines the similarities and differences between G2D2 and continuous methods, particularly in the sampling procedure.
>
> From an algorithmic perspective, G2D2 is similar to DAPS and Resample. These methods alternately perform unconditional inference using a pre-trained diffusion model and fitting to measurement constraints (likelihood terms). Unlike methods such as DPS and PSLD, these methods do not take gradients within the pre-trained model. This is one reason why DPS and PSLD have very high GPU memory consumption. Additionally, in continuous diffusion, the final sample is obtained by gradually denoising and guiding the variables during sampling (i.e., the sample itself is of interest), whereas in G2D2, the variational distribution is the focus. This is an important difference from both an algorithmic and theoretical perspective. We recognize that the recently published survey [3] on inverse problem solving using diffusion models is very useful and believe it will aid in a more detailed comparison.
>
> [3] Daras, et al., "A survey on diffusion models for inverse problems", arXiv preprint, 2024, https://arxiv.org/abs/2410.00083
>
> ### Comparison on motion inverse problem solving (Weakness 3)
>
> We agree that our original submission would benefit from comparative analysis. In response, we have conducted comparisons with existing methods rather than just a demonstration. As comparison methods, we used OmniControl [4], which was a notable work at the time of paper submission, and Guided Motion Diffusion (GMD) [5], which was proposed earlier. Due to time constraints, we have cited the metrics from their papers. We compare the methods in a path following task, specifically, the task of motion generation given a prescribed trajectory for the pelvis. Following OmniControl's setup, we compared FID, R-Precision, Diversity, Foot Skating ratio, Traj. err (50cm), Loc. err (50cm), and Avg. err. within the constraint of 5 frames out of 196 frames on HumanML3D test set.
>
> We present the experimental results in the table below. Details including experimental conditions will be included in the manuscript during the Discussion Period.
>
> | Method      | FID ($\downarrow$)   | R-precision (Top3, $\uparrow$) | Diversity (9.503→) | Foot skating ($\downarrow$) | Traj. Err (50cm) ($\downarrow$) | Loc. err. (50cm) ($\downarrow$) | Avg. err. ($\downarrow$) |
> |-------------|-------|-------------|-------------------|--------------|------------------|------------------|-----------|
> | G2D2        | **0.248** | **0.770**       | 9.381             | **0.048**        | 0.272            | 0.116            | 0.230     |
> | OmniControl | 0.278 | 0.705       | **9.582**             | 0.058        | **0.053**            | **0.015**            | **0.043**     |
> | GMD         | 0.523 | 0.599       | N/A               | 0.086        | 0.176            | 0.049            | 0.139     |
>
> It's important to note that OmniControl and GMD have been fine-tuned for this specific task, while G2D2 is a training-free method. However, in metrics such as trajectory error and location error, OmniControl and GMD show superior performance, which may indicate certain limitations of G2D2 in its current form. We would like to suggest that methods like G2D2 can be used in conjunction with fine-tuned approaches, and that there is still potential for improving G2D2's performance by adopting techniques such as the time-traveling method used in FreeDoM [6].
>
> [4] Xie, et al., "OmniControl: Control Any Joint at Any Time for Human Motion Generation", ICLR2024
>
> [5] Karunratanakul, et al., "GMD: Guided Motion Diffusion for Controllable Human Motion Synthesis", ICCV2023
>
> [6] Yu, et al., "FreeDoM: Training-Free Energy-Guided Conditional Diffusion Model", ICCV2023

---

> > ### Author Response · Authors · 2024-11-23
> > **(Cont.) Rebuttal response to Reviewer ZmJs**
> >
> > ### G2D2 with MaskGIT on image data (Weakness 4)
> >
> > We appreciate this suggestion and have begun integrating MaskGIT as the prior for our image tasks. Initially, we expected results comparable to those obtained with VQ-Diffusion. However, our current results, as shown in the table below, indicate that the performance with MaskGIT is inferior to VQ-Diffusion. We are using the prior model implemented in Pytorch (https://github.com/valeoai/Maskgit-pytorch), which is trained on the ImageNet dataset. This discrepancy might be due to issues with hyperparameters or optimization scheduling, or it could be related to the codebook size of VQ-VAE (4096x128 for VQ-Diffusion and 1024x256 for MaskGIT).
> > | Dataset  | Task                | Prior        | LPIPS | PSNR  |
> > |----------|---------------------|--------------|-------|-------|
> > | ImageNet | Gaussian Deblurring | MaskGIT      | 0.470 | 21.59 |
> > |          |                     | VQ-Diffusion | **0.375** | **22.71** |
> > |          | Super Resolution    | MaskGIT      | 0.462 | 22.04 |
> > |          |                     | VQ-Diffusion | **0.349** | **23.20** |

---

> > > ### Author Response · Authors · 2024-11-27
> > > **Looking forward to your response**
> > >
> > > Thank you for your thorough and insightful review. Although the rebuttal period has been extended, we are eager to address all of your concerns to the best of our ability. We would like to know if you have any remaining questions or issues that we haven't yet addressed. We look forward to your response. Thank you for your time and consideration.

---

> > > > ### Comment · Reviewer_ZmJs · 2024-11-27
> > > >
> > > > I would like to thank the reviewers for their extensive efforts. I believe this is a good paper, and most of my concerns have been resolved. I understand that the main point of the paper is to establish SOTA, and hence I am strongly leaning towards acceptance. Due to some limitations with the results, I will keep my score.

---

### Official Review · Reviewer_eySn · 2024-10-30

**Soundness:** 3
**Presentation:** 4
**Contribution:** 3
**Rating:** 6
**Confidence:** 4

**Summary:**

Authors propose Gradient-guided Discrete Diffusion (G2D2), a novel method for solving linear inverse problems using discrete diffusion as priors. Limitations of discrete diffusion priors are overcame by approximating posterior with variational distribution constructed from categorical distributions and continuous relaxation techniques. Authors demonstrate their method on super-resolution and Gaussian deblurring tasks on ImageNet and FFHQ datasets. G2D2 is compared against popular baselines such as DPS, DDRM, PSLD and ReSample and found to be competitive against continuous diffusion models.

**Strengths:**

* The paper is written very well.
* Using discrete diffusion priors for solving inverse problems is a novel direction that is not well explored.
* Usage of star-shaped noise process is motivated well with big gain in downstream performance compared to the standard Markov noise process.

**Weaknesses:**

* While I appreciate the novelty of using discrete diffusion models in the context of inverse problems, the advantages of using them against continuous counterparts are not motivated well.
* See the questions below.

**Questions:**

* Line 436: "For the image inverse problem experiments, we used text prompts for the VQ-Diffusion model...", do the authors use similar text conditioning for competing methods? If not would it give unfair advantage to G2D2?
* In the limitations section, it is mentioned that "G2D2 does not significantly surpass its continuous counterparts in terms of computational speed or performance". Is there a clear advantage of using discrete diffusions?
* In the appendix, authors provide the hyperparameters used for competing methods (DPS, DDRM, etc.). Are those values taken directly from the corresponding papers or a hyperparemeter search was conducted to find them?
	* If taken directly, I would recommend tuning them separately since in my experience these values are not robust against small changes in the problem setup (some papers add noise to the image in the range [-1,1] some in [0,1] etc.).
	* If the latter one, it would be good to describe which range was searched over how many validation samples.

---

> ### Author Response · Authors · 2024-11-23
> **Rebuttal response to Reviewer eySn**
>
> Thank you for your constructive feedback and for acknowledging our efforts. We will address your questions and concerns below.
>
> ### Motivation for Using Discrete Diffusion Models in Inverse Problem (Weakness 1)
>
> Thank you for your valuable and important comments. Please refer to the "Messages to the Reviewers" for our response on this point.
>
> ### Text Conditioning for VQ-Diffusion Model (Question 1)
>
> We consider this to be an important point. Firstly, in the comparative experiments, we use the setup of comparison methods described in their papers. In PSLD, which uses StableDiffusion v-1.5 as the prior model, prompts are not used (i.e., the model is unconditional), and in ReSample, the model is trained without text conditions. (However, please note that each model is trained on specific datasets.)
>
> We are using text prompts as input for VQ-Diffusion models, but we don't consider this to be an unfair setup. In the ImageNet experiments, we use the general description "a photo of [Class Name]", and for FFHQ experiments, we use "a high-quality headshot of a person". These are not detailed specifications of the target images. Additionally, for reference, we will show results without text conditioning.
>
> | Dataset  | Task                | Text conditioning | LPIPS | PSNR  |
> |----------|---------------------|-------------------|-------|-------|
> | ImageNet | Gaussian Deblurring | Not Used          | 0.410 | 22.21 |
> |          |                     | Used              | **0.375** | **22.71** |
> |          | Super Resolution    | Not Used          | 0.355 | **23.32** |
> |          |                     | Used              | **0.349** | 23.20 |
> | FFHQ     | Gaussian Deblurring | Not Used          | 0.328 | **25.43** |
> |          |                     | Used              | **0.288** | 24.42 |
> |          | Super Resolution    | Not Used          | 0.300 | 26.60 |
> |          |                     | Used              | **0.271** | **26.93** |
>
> From this, it can be observed that text conditioning has a certain effect when using VQ-Diffusion as the prior. This has been added to the revised manuscript.
>
> ### Clear Advantage of using Discrete Diffusions (Question 2)
>
> Please refer to the "Messages to the Reviewers." While the benefits currently include lower GPU memory usage and faster inference when compared, the potential application to multi-modal models could be an additional advantage.
>
> ### Hyperparameter Settings for Comparison Methods (Question 3)
>
> Basically, we have adopted the standard settings for the hyperparameters of these methods as described in their respective papers (corresponding to each dataset).

---

> > ### Comment · Reviewer_eySn · 2024-11-25
> >
> > I would like to thank the authors for their responses and providing results without text conditioning. Even though text prompts are generic (e.g. "a photo of {class_name}"), experiments show the clear performance boost of using such text prompts. While comparison methods may have stronger diffusion backbones, I don't think that including text prompts make it a fair comparison.
> >
> > Overall, I appreciate the novelty of using discrete diffusion models to solve inverse problems but still have doubts around its benefits compared to continuous counterparts (less GPU usage is not convincing since DDRM has comparable GPU usage but much shorter inference time and comparable performance) based only on the current results. For these reasons I will maintain my score.

---

### Official Review · Reviewer_TwFi · 2024-11-04

**Soundness:** 3
**Presentation:** 3
**Contribution:** 1
**Rating:** 5
**Confidence:** 3

**Summary:**

The authors propose a method for solving inverse problems using discrete diffusion models with gradient guidance. They utilize the Gumbel trick to relax the categorical distribution, allowing for gradient computation. Additionally, they introduce star-shaped diffusion models to address the limitations of conventional discrete diffusion models. This approach increases the probability of transitioning to the absorbing state, enabling correction of erroneous codes. In experiments, the proposed method outperforms previous continuous pixel/latent domain diffusion model-based inverse solvers in super-resolution and deblurring tasks. They also apply it to a path-following task using a generative masked motion model.

**Strengths:**

The authors propose novel methods to address challenges arising when adapting previous approaches to the new task of discrete inverse problems.

**Weaknesses:**

The motivation for using a discrete diffusion model in the image domain is lacking. Given the availability of options to reduce computational burden, such as low-precision floating points or lighter models, the necessity of a discrete representation is not clearly justified. It would be more convincing if the proposed method demonstrated its advantages in areas where discrete representations are essential, such as language or molecular modeling. To strengthen the paper, consider the following suggestions:
- Compare the discrete approach directly with continuous models that use low-precision or lightweight architectures.
- Include experiments in domains where discrete representations are inherently suitable, such as text generation or molecular modeling.
- Discuss any potential advantages of discrete representations in image domains that may not be immediately clear from the current results.

**Questions:**

- How does the discrete star-shaped diffusion process compare with its continuous counterparts in terms of similarities and differences? Could you discuss the specific challenges the authors faced when adapting star-shaped diffusion to discrete spaces and the strategies used to address them? Additionally, please elaborate on any unexpected advantages or limitations of the discrete version compared to the continuous model.
- Could discrete star-shaped diffusion models be integrated with diffusion model-based solvers like DDRM or DDNM?
- The proposed method outperforms even continuous-domain algorithms in image tasks. Were the experiments conducted under computational constraints? If so, please specify these restrictions and provide evaluations under unrestricted conditions as well.

Possible Errors
- Proof of Lemma B.3: The proof appears incorrect. In Lines 975 to 980, the claim that the reverse transition inverts the forward process and leads to equality with $\delta_{z_0, z_0{\prime}}$ should be revised. While the lemma’s conclusion is unaffected, replace exact equalities with equality in distribution where appropriate.
- Line 265: $z_t \to z_{t-1}$.

---

> ### Author Response · Authors · 2024-11-23
> **Rebuttal response to Reviewer TwFi**
>
> We appreciate your thoughtful feedback and recognition of our work. Below, we provide answers to your questions and address your concerns.
>
> ### Motivation for Using Discrete Diffusion Models in the Image Domain is Lacking (Weakness 1)
>
> Please refer to the "message to the Reviewers" comment. Your points are valid, and we address each of your detailed suggestions as follows.
>
> ### Comparison with low-precision or lightweight continuous models (Weakness 2)
>
> We acknowledge that comparing our work with papers such as "Q-DM: An Efficient Low-bit Quantized Diffusion Model" could be considered. Nonetheless, it is important to note that the application of such models to inverse problem settings has not yet been explored. This could be a valuable direction for future research.
>
> ### Include experiments in domains where discrete representations are inherently suitable (Weakness 3)
>
> Given the time constraints and considering the scope of this paper, it is currently challenging to apply G2D2 to molecular or text data. However, we would like to point out that the use of prior models is also being discussed in these fields.
>
> ### Discuss any potential advantages of discrete representations in image domains (Weakness 4)
>
> As mentioned in the General comments, we believe that our method is compatible with approaches like Transfusion and Show-o, which treat image data as tokens in the same space as text data and train them using a single transformer model. We believe this compatibility could potentially give our method an advantage over continuous counterparts in the future.
>
> ### Comparison and Adaptation Challenges of Discrete Star-Shaped Diffusion Process vs. Continuous Models (Question 1)
>
> We share a similarity with the DAPS paper in that we introduced a star-shaped noise process in the context of solving inverse problems, just as they did in the continuous version. However, among the benefits obtained from this introduction, the advantage of enabling "remasking" is unique to discrete diffusion models with mask-absorbing states. This brings interpretable benefits to this problem setting. The effect is evident when looking at Tables 1 and 2.
>
> ### Integration of Discrete Star-Shaped Diffusion Models with Diffusion Model-Based Solvers like DDRM or DDNM (Question 2)
>
> This is a very interesting discussion, but I believe it's difficult to naturally integrate with methods using pixel-domain diffusion models such as DDRM and DDNM. The main reason is that these methods assume linearity between the output domain of the diffusion model and the measurement domain, and further assume that the singular value decomposition (SVD) of this linear operator can be fully obtained. While these methods are indeed effective when these assumptions are met, they simultaneously limit the usable priors and the problem settings that can be addressed.
>
> ### Impacts of Computational constraints (Question 3)
>
> In our experiments, we did not impose any restrictions on the computational complexity of each method. The number of time steps and other factors that determine computational complexity were adopted from the standard settings described in each respective paper. Details on GPU memory usage and execution time (wall-clock time) for each setting are provided in Appendix C.5.
>
> ### Possible Errors
>
> Thank you for your feedback. In the revised manuscript, we have corrected the proof of Lemma B. 3. This proof assumes that in the discrete diffusion model, \( z_{T} \) has completely lost the information of \( z_{0} \).

---

> > ### Author Response · Authors · 2024-11-27
> > **Looking forward to your response**
> >
> > Thank you for your thorough and insightful review. Although the rebuttal period has been extended, we are eager to address all of your concerns to the best of our ability. We would like to know if you have any remaining questions or issues that we haven't yet addressed. We look forward to your response. Thank you for your time and consideration.

---

> ### Comment · Reviewer_TwFi · 2024-12-02
>
> Thank you for addressing the reviewers’ concerns.
>
> However, there is still an issue with the revised proof. Specifically, $q(x_T|x_0)\neq q(x_T)$ because the finite Markov chain does not fully erase the information. Therefore,  $q(x_T|y)$  is only approximately constant. Since this discrepancy $q(x_T) \neq q_{\text{sample}}(x_T)$ ($\equiv Constant$) is a critical aspect of diffusion modeling, it should be explicitly addressed in the main text.
>
> Furthermore, I am still uncertain about the significance of this work. While discrete diffusion models and inverse problems are both important topics, solving inverse problems using discrete diffusion models does not seem particularly meaningful. Considering that most important inverse problems are inherently continuous in nature, the motivation for focusing on discrete settings remains unclear.

---

> > ### Author Response · Authors · 2024-12-03
> >
> > We would like to express our sincere gratitude for your time in reviewing the Rebuttal for our paper and for providing your valuable comments. We are fully committed to improving our paper to the best of our ability within the given time constraints. Your feedback is greatly appreciated and will significantly contribute to enhancing our work.
> >
> > We appreciate your pointing out the issue with the current proof. We agree with your comment that in general mask absorbing state discrete diffusion, while $q(x_T|x_0)$ approaches the Prior $q(x_T)$, this is not completely achieved depending on the parameter settings. (However, in practical cases, as described in Appendix C.2, in the VQ-Diffusion setting we use, the elements of $z_T$ become [MASK] tokens with a probability of 99.999%, which is why we stated this assumption in the main text and used it in the proof.)
> >
> > After your comment and upon further consideration, we have decided to proceed with the proof of Lemma B.3 without relying on the assumption that $q(z_T|z_0)$ (or $q(z_T|y)$) coincides with $q(z_T)$. As the period for modifying the main text has already concluded, we will provide an outline of this revised proof here.
> >
> > First, the conditional distribution $q_{sampling}(z_T|y)$ in $q_{sampling}$ is identical to the conditional distribution $q_{star}(z_T|y)$ of the star-shaped noise process. At this point, both distributions are exactly the same.
> >
> > From here on, we can show that for each $q_{sampling}(z_t|y)$, starting from $z_T$ and proceeding in the order $z_{T-1}, ..., z_t, ..., z_0$, it matches $q_{star}(z_t|y)$ as a marginal distribution. This can be confirmed in terms of marginal distributions, given that $q_{sampling}$ uses the conditional distribution of the star-shaped noise process. For example, when $z_{t+1} \sim q_{sampling}(z_{t+1}|y) = q_{star}(z_{t+1}|y)$, we have $z_t \sim E_{z_{t+1} \sim q_{sampling}(z_{t+1}|y)}[q_{sampling}(z_t|z_{t+1}, y)] = E_{z_{t+1} \sim q_{star}(z_{t+1}|y)}[q_{star}(z_t|z_{t+1}, y)] = q_{star}(z_t|y)$.
> >
> > It is important to note that $q_{sampling}$ and $q_{star}$ match only when we focus on each $z_t$ and marginalize over the other variables $(z_{0:t-1}, z_{t+1:T})$. G2D2 enables posterior sampling by omitting the complex dependencies of $z_{1:T}$.

---

> > > ### Author Response · Authors · 2024-12-03
> > >
> > > > Furthermore, I am still uncertain about the significance of this work. While discrete diffusion models and inverse problems are both important topics, solving inverse problems using discrete diffusion models does not seem particularly meaningful. Considering that most important inverse problems are inherently continuous in nature, the motivation for focusing on discrete settings remains unclear.
> > >
> > > We appreciate your feedback and understand your concerns regarding the significance of focusing on discrete diffusion models for inverse problems. However, recent trends show that discrete (or masked) diffusion models are achieving remarkable results across various modalities, sometimes outperforming continuous models. Here are some examples:
> > >
> > > - **Motion synthesis**: [Guo, 2024] and [Pinyoanuntapong, 2023]
> > > - **Text-to-image (T2I)**: [Weber, 2024]
> > > - **Text-to-music (T2Music)**: [Ziv, 2024]
> > >
> > > These models excel in generating high-fidelity outputs but often face limitations in controllable generation, particularly for inverse problem applications.
> > >
> > > Our work addresses a critical gap by proposing a framework that integrates these discrete diffusion models into inverse problem-solving. This not only expands their applicability but also leverages their unique advantages to tackle a broader range of challenges. We believe this exploration is both timely and significant, given the rising prominence of discrete diffusion models in generative tasks.
> > >
> > > Furthermore, as mentioned in our "Messages to Reviewers,", when considering the potential benefits of discrete diffusion models in inverse problem solving, recent studies such as **Transfusion** [1] and **Show-o** [2] might expand the possibilities in this field. These models enable the generation of multimodal data (text and images) using a common discrete token space and Transformer model, which we believe could be compatible with our approach. By incorporating similar principles, our framework could potentially broaden its scope to multimodal inverse problems, aligning with recent advances in generative modeling.
> > >
> > > Once again, thank you for your valuable feedback and the time you've invested in reviewing our paper. We appreciate your insights and are open to further discussion as time permits. Your comments are helping us improve our research, and we're grateful for your contribution to advancing the field.
> > >
> > > [Guo, 2024], “MoMask: Generative Masked Modeling of 3D Human Motions”, CVPR2024
> > >
> > > [Pinyoanuntapong, 2024], “MMM: Generative Masked Motion Model”, CVPR2024
> > >
> > > [Weber, 2024] “MaskBit: Embedding-free Image Generation via Bit Tokens”, 2024
> > >
> > > [Ziv, 2024]  "Masked audio generation using a single non-autoregressive transformer”, ICLR2024

---

### Author Response · Authors · 2024-11-23
**Message to the Reviewers**

Dear Reviewers,

We would like to express our sincere gratitude for your thorough review of our paper. We have carefully considered each of your valuable comments and questions.

## Revisions

Based on the feedback we received, we have made the following revisions to our paper. Please find below a summary of the modifications:

1. In the sections explaining our theoretical framework, we corrected typographical errors and rewrote the content to make it clearer and more easily understandable.
2. The proof of Lemma B.3 in the Appendix has been corrected.
3. We have added several references to practical applications in the Related Works section of the Appendix.
4. As additional experiments in the Appendix C.7, we have included the results of our investigation into the impact of text conditioning on performance.
5. We have added an analysis of failure modes to Appendix C.8.
6. We updated the notation in the graphical model figure.

## Response to Key Feedback

We received feedback from our reviewers, primarily pointing out that "The motivation for using discrete diffusion models in the image domain is lacking." We consider this observation to be extremely important for exploring this topic. We would like to address this point as follows:

1. It is acknowledged that this research is the first paper to use discrete diffusion models for solving image inverse problems. However, it may be difficult to demonstrate significant advantages of using discrete diffusion models based solely on the current results.

2. Nevertheless, it is noteworthy that we were able to show competitive results with less GPU memory usage and faster inference compared to similar continuous space methods like PSLD and Resample, in terms of applying diffusion models on the learned latent space. Additionally, since performance is also influenced by the quality of pre-trained models, we may expect further performance improvements if better models than VQ-Diffusion become available in the future.

3. Furthermore, when considering the potential benefits of discrete diffusion models in inverse problem solving, recent studies such as Transfusion [1] and Show-o [2] might expand the possibilities in this field. These models enable the generation of multimodal data (text and images) using a common discrete token space and Transformer model, which we believe could be compatible with our approach.

## References

[1] Zhou et al., "Transfusion: Predict the next token and diffuse images with one multi-modal model", arXiv preprint, 2024, https://arxiv.org/abs/2408.11039

[2] Xie et al., "Show-o: One single transformer to unify multimodal understanding and generation", arXiv preprint, 2024, https://arxiv.org/abs/2408.12528

---

### Meta-Review · Area_Chair_Jnh8 · 2024-12-19

**Metareview:**

The main claim of the work is to be the first approach to use discrete diffusion model-based priors for solving image inverse problems.

While this is an interesting direction, and the paper is solid (and minor mistakes were fixed in the reviewer discussion phase), the motivation of using discrete diffusion models in the image domain still remains unclear. Empirical results presented in the paper are lagging behind the state-of-the-art. If other domains like molecular generation or motion modeling are potential candidates for this method, the work will benefit from a major revision with another round of reviews to make this point more appreciable.

I recommend to reject the paper at this point and encourage the authors to take the reviewers feedback into account to make the motivation of the paper more appealing.

**Additional Comments On Reviewer Discussion:**

Various points were raised during the rebuttal phase, such as errors in a proof of Lemma B.3. These were all addressed in the rebuttal phase. However, reviewers TwFi and eySn still found the motivation of discrete models for continuous imaging inverse problems to be questionable. I agree with these points, and believe the paper will benefit from another round of reviews to explain the motivation more clearly.

---

### Decision · Program_Chairs · 2025-01-22

Reject